# Cellular encoding of Cy dyes for single-molecule imaging

Lilia Leisle, Rahul Chadda, John D Lueck, Daniel T Infield, Jason D Galpin, Venkatramanan Krishnamani, Janice L Robertson, Christopher A Ahern*

Department of Molecular Physiology and Biophysics, University of Iowa Carver College of Medicine, Iowa City, United States

**Abstract** A general method is described for the site-specific genetic encoding of cyanine dyes as non-canonical amino acids (Cy-ncAAs) into proteins. The approach relies on an improved technique for nonsense suppression with in vitro misacylated orthogonal tRNA. The data show that Cy-ncAAs (based on Cy3 and Cy5) are tolerated by the eukaryotic ribosome in cell-free and whole-cell environments and can be incorporated into soluble and membrane proteins. In the context of the *Xenopus laevis* oocyte expression system, this technique yields ion channels with encoded Cy-ncAAs that are trafficked to the plasma membrane where they display robust function and distinct fluorescent signals as detected by TIRF microscopy. This is the first demonstration of an encoded cyanine dye as a ncAA in a eukaryotic expression system and opens the door for the analysis of proteins with single-molecule resolution in a cellular environment.

**\*For correspondence:**
christopher-ahern@uiowa.edu

**Competing interests:** The authors declare that no competing interests exist.

## Introduction

Fluorescent reporters are useful for the study of protein dynamics in a live cell, because they can inform on the conformational dynamics of a protein. A common strategy to incorporate fluorophores into target proteins is the fusion to fluorescent proteins, such as GFP or one of its spectral variants (*Miranda et al., 2013*; *Zachariassen et al., 2016*). This approach has the built-in convenience of encoding but their positioning within the target protein can be limited by their large size, which may impact protein function or trafficking. Alternatively, post-translational chemical labeling with compact fluorescent dyes through reactive side-chains (Cys, Lys) or with bio-orthogonal labeling allows for the use of more diverse fluorophores and experimental applications (*Mannuzzu et al., 1996*; *Cha and Bezanilla, 1997*; *Priest et al., 2015*; *Debets et al., 2013*). Nonetheless, Cys- or Lys-labeling in live cells is limited to extracellular residues and often results in a substantial background signal due to native reactive residues of the cell. This shortcoming is further limiting when labelling side-chains in eukaryotic membrane proteins which can have many 'background' cysteine residues, that when mutated can lower expression profiles or have functional consequences. Bio-orthogonal labeling is advantageous in this respect (i.e. reduced off-target labeling, increased specificity) but requires that the residues or regions for modification are chemically accessible. Fluorescent non-canonical amino acids (ncAA) present a solution to these drawbacks as they are relatively small, and theoretically, can be encoded at any site within the protein in cellular and cell-free environments (*Turcatti et al., 1996*; *Cohen et al., 2002*; *Zhang et al., 2004*; *Summerer et al., 2006*; *Wang et al., 2006*; *Kajihara et al., 2006*; *Pantoja et al., 2009*; *Kalstrup and Blunck, 2013*). However, of the available encodable fluorophores, most are excited by near UV light, a spectroscopic property that leads to competing cellular fluorescence, and have limited, if any, utility for single-molecule studies e.g. they have very short fluorescence lifetimes.

Further, ongoing advances in protein structure determination are revealing the macromolecular structures of membrane protein complexes and in some cases have started to provide mechanistic

**eLife digest** Many scientists would argue that the leading edge of biological exploration is playing out at the level of individual molecules. On this scale, the essential molecular players of life are so small that they simply cannot be seen with a normal light microscope. While technology that can capture static snapshots of individual proteins frozen in time continues to advance, the choice of tools to observe individual proteins in action remains limited. Moreover, each of the existing tools for studying protein dynamics in living cells also has its own caveats.

These issues led Leisle et al. to set out to develop a new method that would allow researchers to study individual proteins in live cells. This goal required a probe that was relatively small, bright, stable and compatible with biological samples. Fluorescent probes called "Cy dyes" meet all these criteria. Leisle et al. turned these probes into amino acids, the building blocks of proteins, and then supplied them to cells that were genetically programmed to incorporate the probes into a protein of interest as it was being built. This new technique allows the protein to be marked at specific sites and stops other proteins from being labeled by mistake (a common problem with other protein labeling methods).

This new approach was confirmed to work, firstly, in a cell extract and, secondly, in an intact cell with two unrelated proteins found in the cell membrane. The cells used were frog eggs, a type of cell that is widely used in biological experiments. Yet this approach should be easy to apply to any protein and, in theory, to any cell type after it has been optimized. The next challenges include finding ways to get the probe incorporated more efficiently into the protein of interest and to protect the probes from losing their brightness – a phenomenon called quenching.

Finally, studies of single molecules provide the deepest possible insight into how a protein in a living cell carries out its activities. Better tools for single-molecule studies will lead to a more complete understanding of the dynamic life of proteins in action. Moreover, in the case for those proteins that are related to diseases, these kinds of studies may in future guide the development of new or improved drugs to treat disease.

glimpses into their function (*Eisenstein, 2016*). However, a structure represents a snapshot of a single conformation, yet proteins constantly cycle though various states, all of which have the potential to be functionally significant. Therefore, although high-resolution protein structures are becoming more commonplace, assigning functional correlates remains a significant challenge. This is true especially for eukaryotic membrane proteins, many of which cannot be easily produced in biochemical scales and for which expression in a live cell is a prerequisite, e.g. as is the case for many voltage-gated plasma-membrane channels. Thus, there is a growing general need for improved encoded fluorescent reporters, ideally those that are well suited to single-molecule studies and are applicable to eukaryotic expression systems.

Herein a method is described for genetic encoding of organic Cy dyes as non-canonical amino acids (Cy-ncAAs), in cell-free and cellular environments. We first report a synthetic approach for the production of orthogonal misacylated tRNAs that carry a Cy-ncAA. The tolerance of the eukaryotic ribosome for these fluorophores is demonstrated in a eukaryotic cell-free protein expression system with a novel luciferase rescue assay. We then show in Xenopus oocytes that these Cy-dye-based optical probes (Cy3, Cy5 and the self-healing Cy3 variant LD550) can be incorporated into membrane proteins (providing two examples: a chloride ion channel and a voltage-gated sodium channel auxiliary subunit). These proteins were properly folded, functional and trafficked to the plasma membrane, albeit in reduced levels compared to wild-type channels. TIRF microscopy was then used to obtain single-molecule images from cells expressing membrane proteins containing Cy3 and Cy5, further confirming that the encoding is successful.

## Results and discussion

A method is presented for the cellular encoding of single-molecule fluorophores as non-canonical amino acids based on widely employed organic Cy dyes for single-molecule studies (*Figure 1*). These encoded Cy-ncAAs were produced through a hybrid strategy based on the design elements

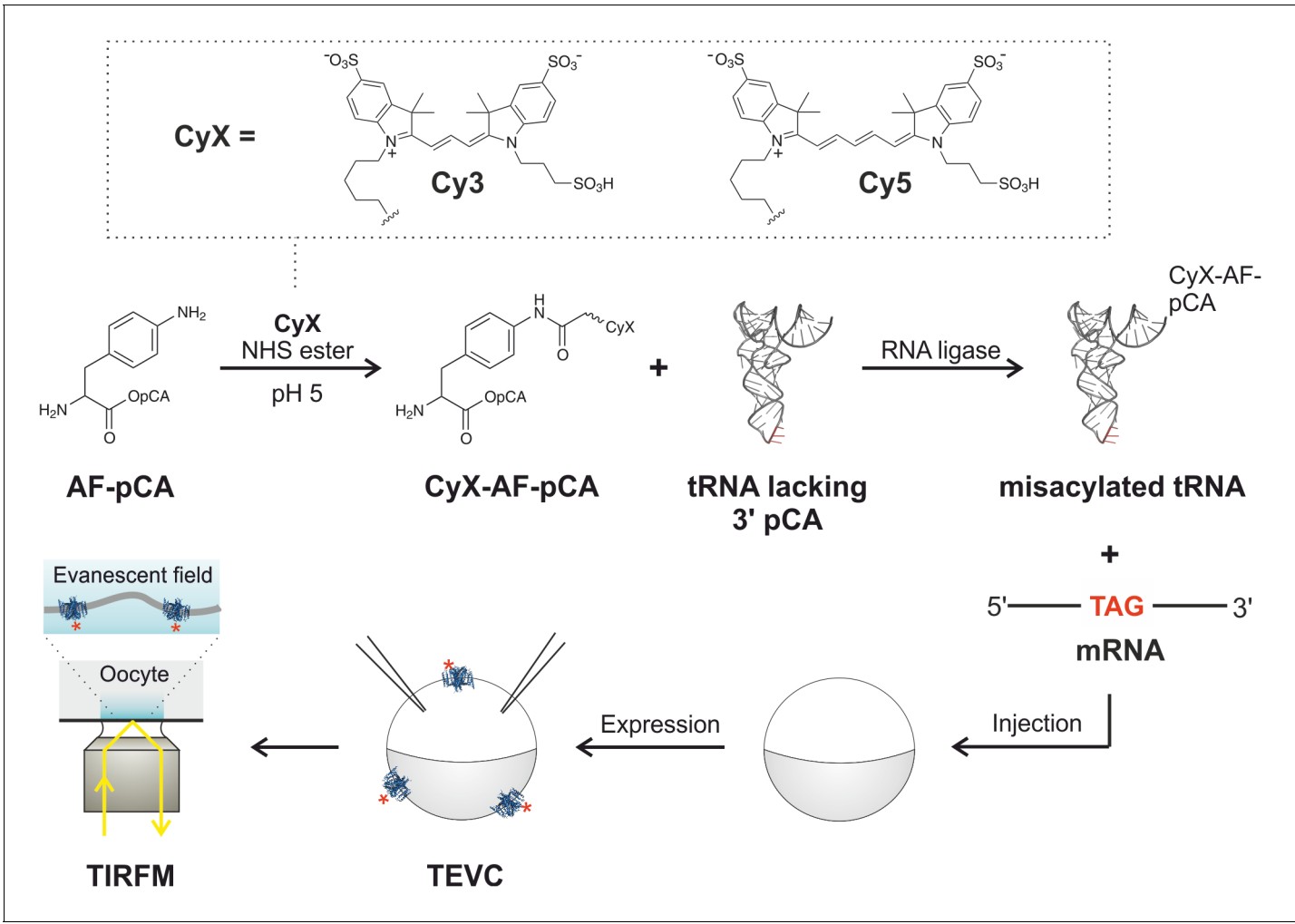

**Figure 1.** Experimental flow for synthesis and incorporation of Cy fluorophores as ncAAs for single-molecule imaging of plasma membrane proteins in *X. laevis* oocytes. Dashed box (top) shows fluorophores used in this study. For details of the procedure see main text.

The following figure supplements are available for figure 1:

**Figure supplement 1.** HPLC chromatogram (**A**), UV absorbance spectrum (**B**) and LRMS (**C**) of AF-pCA.

**Figure supplement 2.** HPLC chromatogram (**A**), UV absorbance spectrum (**B**) and LRMS (**C**) of Cy3-AF-pCA.

**Figure supplement 3.** HPLC chromatogram (**A**), UV absorbance spectrum (**B**) and LRMS (**C**) of Cy5-AF-pCA.

**Figure supplement 4.** HPLC chromatogram (**A**) and UV absorbance spectrum (**B**) of LD550-AF-pCA.

of misacylated tRNA for cell-free expression systems (*Kajihara et al., 2006*) and the genetic code expansion technique of nonsense suppression in *Xenopus laevis* oocytes (*Leisle et al., 2015*). Briefly, a free primary amine containing amino acid, such as para-amino-L-phenylalanine (AF), is chemically conjugated to the dinucleotide phospho-desoxy-cytosine phospho-adenosine (short pCA) and subsequently reacted with a commercially available Cy dye succinimide ester under acidic conditions to ensure selective labeling of the para-amino group (*Figure 1*; *Figure 1—figure supplements 1–4*). This dinucleotide-ncAA product is HPLC purified and then enzymatically coupled in vitro to an orthogonal tRNA which is competent to suppress an in-frame nonsense codon in a cell-free translation reaction or in the context of the *Xenopus* oocyte, as shown in *Figure 1*. The latter allows for electrophysiological determination of ion channel function at the plasma membrane by two-

electrode voltage clamp (TEVC) recordings, and oocytes are an established experimental platform to visualize single ion channel complexes using total internal reflection fluorescence (TIRF) microscopy (*Ulbrich and Isacoff, 2007*; *Sonnleitner et al., 2002*).

The fluorescence of the Cy-ncAAs provided a straightforward way to assess the tRNA ligation efficiency through the measurement of relative absorbance of the nucleic acid and the cyanine chromophore of an acylated tRNA or by HPLC. Surprisingly, the standard ligation conditions (37°C, 40 min) used for nonsense suppression by our lab and others (*Pantoja et al., 2009*; *Pless et al., 2013*, *2014*; *Nowak et al., 1998*) resulted in relatively poor tRNA acylation yields with Cy-ncAAs (*Figure 2A*). We reasoned that this low ligation efficiency could, in part, be due to hydrolysis of the Cy-ncAA from the tRNA occurring during the enzymatic ligation reaction at 37°C; hydrolysis is a highly temperature sensitive process (*Stepanov and Nyborg, 2002*). This possibility was tested by performing the misacylation reaction at a lower temperature (4°C). The incubation time was increased (to 2, 5 or 11 hr) to overcome the reduced enzymatic activity of the RNA ligase induced by the lower temperature. This approach identified conditions at 4°C for 2 hr with higher ligation efficiencies: $68 \pm 10.2\%$ for Cy3-ncAA and $79 \pm 2.5\%$ for Cy5-ncAA (*Figure 2A*). The results for ligations at 4°C, 5 hr were not significantly different from samples obtained after 2 hr (Cy3: p=0.76; Cy5:

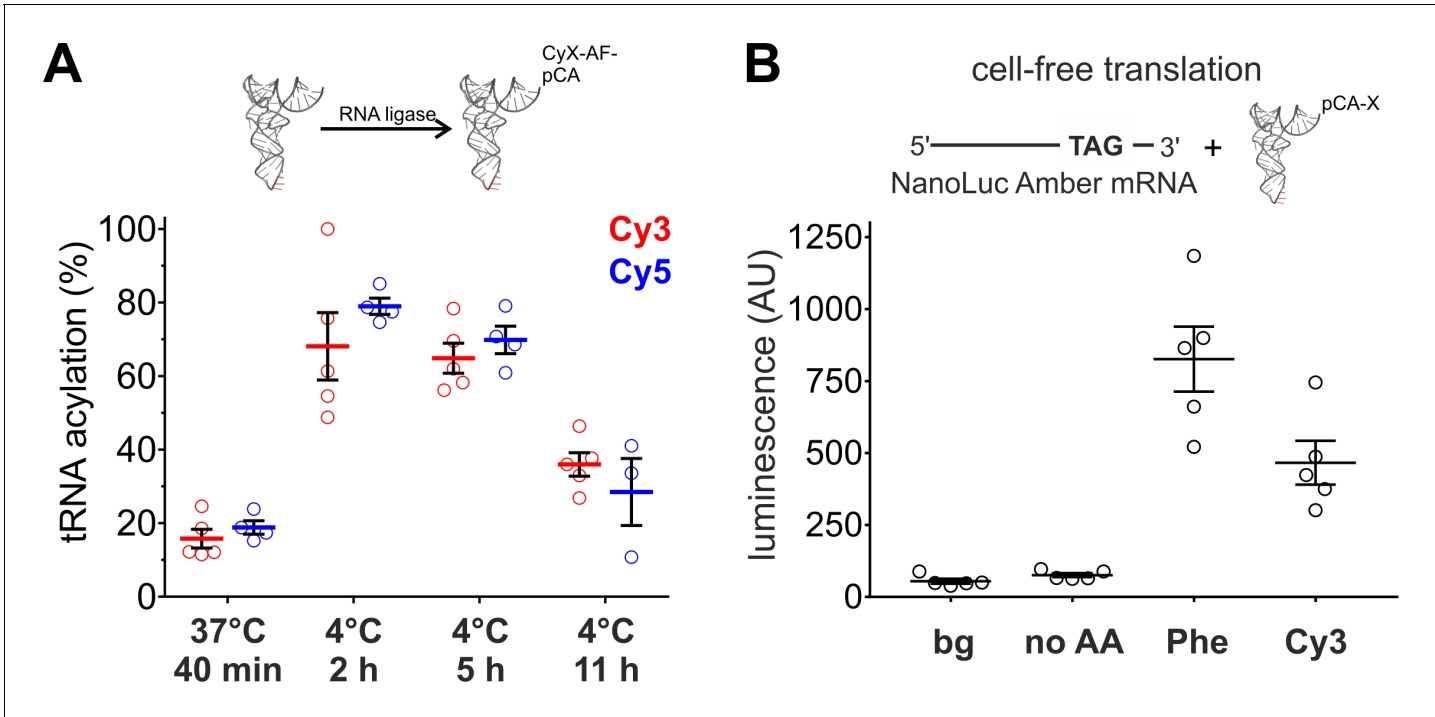

**Figure 2.** Optimization of tRNA acylation and genetically encoding Cy-ncAAs into soluble proteins in a eukaryotic cell-free expression system. (**A**) In vitro acylation efficiency of tRNA with Cy3- or Cy5-AF was determined by absorbance measurements for different RNA ligation conditions (Cy3: N = 5; Cy5: N ≥ 3). Highest efficiencies were found at 4°C after 2 or 5 hr. 4°C, 2 hr was chosen as optimal reaction condition for all further experiments. (**B**) In a eukaryotic in vitro translation system, Luciferase activity of NanoLuc Amber was rescued by Cy3-AF-tRNA produced 56% of the Phe suppression signal (826 ± 126 AU versus 466 ± 85 AU for Phe and Cy3, respectively). Both of which were well above background (bg) signal or from the Nanoluc Amber plus empty tRNA (no AA). All were quantified by bioluminescence (AU, arbitrary units). The read-through control (no AA) was not significantly different from the background luminescence (bg; p>0.1). NanoLuc Amber Phe is ~11 fold above the no AA condition and Cy3-AF ~6 fold (N = 5, individual values with mean ± s.e.m.).

The following figure supplements are available for figure 2:

**Figure supplement 1.** HPLC purification of acylated pyrrolysine tRNA (PylT).

**Figure supplement 2.** In eukaryotic cell-free translation system, the yield of nonsense suppressed NanoLuc Amber (Phe, N = 5, same data as in *Figure 2B*) is roughly 7000 times lower than for wild-type NanoLuc (WT, N = 3).

p=0.08) and the acylation efficiency decreased significantly after an incubation time of 11 hr. Consequently, for the following experiments, all tRNA acylation reactions were done at 4°C for 2 hr. These acylation yields were also confirmed by HPLC where Cy3 and Cy5 ligation reactions were similar and displayed high (>80%) acylated yields (*Figure 2—figure supplement 1*). However, the HPLC purified tRNA fraction diminished the yield, roughly 10% of the starting material, resulting in amounts that were insufficient for proper experimental validation. Hence, non-HPLC purified aminoacylated tRNA were employed for all subsequent non-sense suppression experiments.

To determine if large amino acids like the Cy-ncAAs could be tolerated by the eukaryotic ribosome, they were encoded into target proteins in a rabbit reticulocyte cell-free protein expression system (*Figure 2B*). An amber (TAG) version of the high-activity NanoLuc, a small luciferase subunit from the deep-sea shrimp *Oplophorus gracilirostris* (*Hall et al., 2012*), was generated by inserting the amber stop codon TAG between Gyl159 and Val160 and named NanoLuc Amber. A pyrrolysine tRNA from *Methanosarcina mazei* (PylT) was chosen for nonsense suppression given that it has been shown to be orthogonal in eukaryotic cells (*Hall et al., 2012*; *Mukai et al., 2008*); however, to our knowledge, this is the first use in a cell-free protein synthesis system in an in vitro misacylated form. Applying the optimized tRNA ligation conditions, PylT was employed to encode Cy3-AF or Cy5-AF or the natural amino acid Phe within NanoLuc Amber. Rescue of the NanoLuc Amber full length protein via TAG suppression was quantified in a luminescence assay (*Figure 2B*). Eukaryotic protein synthesis reactions supplemented with NanoLuc Amber cRNA in the presence of PylT lacking an appended amino acid (no AA) produced no measurable bioluminescence above the background signal (p>0.1), consistent with a truncated, nonfunctional NanoLuc Amber protein, and demonstrates orthogonality of the Pyl tRNA (*Figure 2B*). In contrast, supplementing the in vitro translation reactions with NanoLuc Amber cRNA and PylT carrying Phe or Cy3-AF resulted in NanoLuc Amber rescue as shown by bioluminescence (*Figure 2B*). Here, Cy3 incorporation resulted in 56% of the expression of Phe suppression (826 ± 126 AU versus 466 ± 85 AU for Phe and Cy3, respectively). Notably, in the cell-free production, NanoLuc amber rescue was poor for Phe and Cy3, which had yields roughly 7000 and 13,000-fold, respectively, lower than wild-type NanoLuc (*Figure 2—figure supplement 2*). These data indicate that large fluorescent amino acids based on Cy-dyes can be encoded by the eukaryotic translation machinery through nonsense suppression.

We next tested the possibility that Cy-ncAAs could be encoded in the cellular environment of the Xenopus oocyte expression system given its ease for translation of cRNA transcripts and nonsense suppression with in vitro misacylated tRNA in combination with electrophysiological and single-molecule examination (*Leisle et al., 2015*; *Dougherty and Van Arnam, 2014*). The tRNA variant most often used for amber codon suppression in oocytes is THG73 (*Tetrahymena thermophila* G73) (*Saks et al., 1996*). The chloride channel ClC-0 was chosen as a representative membrane protein as previous experiments on ClC-0 have shown that the gating glutamate (E166), a transmembrane residue in the ion permeation pathway, is widely tolerant to natural amino acid substitutions, including large-volume side chains like Phenylalanine, while showing robust expression of chloride conducting ion channels (*Zhang et al., 2009*). Available crystal structures of gating glutamate mutants of EcClC (a prokaryotic CLC homologue) suggest that substitutions at this position cause the side chain to orient towards the extracellular space with access to aqueous solution, resulting in an open pore phenotype (*Dutzler et al., 2003*; *Lobet and Dutzler, 2006*). This position also turned out to be amenable to nonsense codon suppression as suggested by rescued channel function of ClC-0 E166TAG by Phe-tRNA (*Figure 3A–C*). To test that the rescued currents were mediated by ClC-0, chloride was substituted by iodide, which has been shown to inhibit ClC-0 currents (*Pusch et al., 1995*; *Ludewig et al., 1997*; *Zifarelli and Pusch, 2007*), resulting in a strong reduction of the measured current amplitudes (*Figure 3—figure supplement 1*). Phenylalanine was used as a positive control for nonsense suppression of ClC-0 E166TAG because Phe is a close mimic of the core structure of the Cy-ncAA. However, we also note that under our current experimental conditions, the wild-type glutamate amino acid was not successfully coupled to pCA, barring its use for nonsense suppression. The 'read-through' at the introduced TAG codon in position 166, a spurious process in genetic code expansion techniques, was minimal at 24 hr post-injection in the presence of a nonacylated (no AA) tRNA (*Figure 3A–C*). However, these bleed-through currents generally increase with time, in a site and channel specific manner, and this was also the case with E166TAG ClC-0 expression past 24 hr. Further, max. 7.5 ng of ClC-0 E166TAG cRNA was used per oocyte as higher cRNA amounts led to spurious but increased read-through expression. Consistent with results obtained in

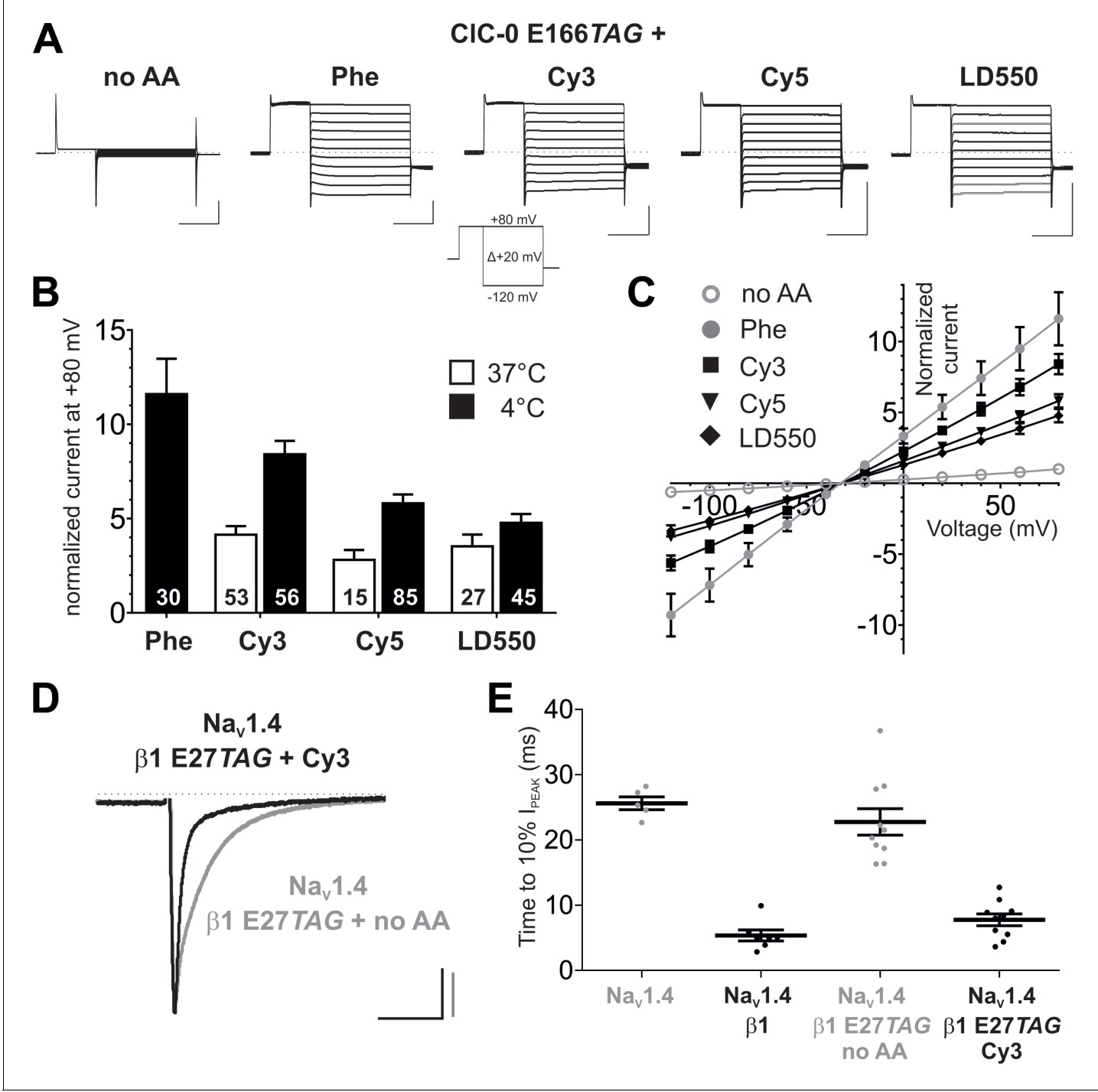

**Figure 3.** Genetic encoding of Cy-ncAAs into membrane proteins in *Xenopus laevis* oocytes. (**A**) to (**C**) Nonsense suppression of ClC-0 E166*TAG*. (**A**) The chloride channel function of ClC-0 was successfully reconstituted with the misacylated tRNAs (Phe, Cy3, Cy5, LD550) while nonacylated tRNA (no AA) yielded no functional channels. Representative TEVC current recordings and the voltage clamp protocol are shown. Horizontal scale bars indicate time (50 ms), vertical scale bars the current amplitude (10 μA). Note that traces were scaled to unity for clearer presentation of current properties. (**B**) For quantification, currents elicited by a +80 mV pulse in oocytes expressing reconstituted chloride channels were normalized to currents of the background control (no AA). Black bars indicate rescue by suppressor tRNA aylated at 4°C for 2 hr, white bars show results for tRNA acylated at 37°C for 40 min. For Cy3 and Cy5, rescue of ClC-0 E166*TAG* has been significantly increased by using tRNAs acylated at 4°C (Cy3: p<0.001; Cy5: p=0.01; LD550: p=0.11). Values for the fold increase in conductance are shown as mean ± s.e.m.; numbers of individual oocytes tested are indicated (for 4°C results were pooled from 5–11 different batches of oocytes; for 37°C from 2–7 batches). (**C**) Steady-state current-voltage relationships for all conditions shown in (**A**) and recordings quantified in (**B**) for 4°C. As predicted, the reconstituted channels behave as voltage-independent, constitutively open

*Figure 3 continued on next page*

Figure 3 continued

conductances with a reversal potential around the chloride equilibrium potential, indicative of a chloride-selective channel. (D) and (E) Nonsense suppression of Na$_V$β1 E27*TAG*. (D) Representative TEVC recordings of oocytes coinjected with Na$_V$1.4 and β1 E27*TAG* + Cy3-tRNA (black) or β1 E27*TAG* + nonacylated tRNA (grey). Accelerated Na$_V$1.4 fast-inactivation kinetics (black) indicate successful incorporation of Cy3 into β1. Currents were elicited by a 50 ms test pulse to −20 mV from a holding potential of −120 mV. They were scaled to unity for illustration purposes; scale bars: horizontal indicates time (20 ms), vertical indicates current amplitude (1 μA). (E) For quantification of the effect on inactivation kinetics the time period till the peak current (I$_{PEAK}$) decayed to 10% of its initial value was estimated. Results for Na$_V$1.4 + β1 E27*TAG* + no AA (23 ± 2.1 ms) were not significantly different from Na$_V$1.4 expressed alone (26 ± 1.1 ms; p=0.36). Both, wild-type β1 and the reconstituted β1, significantly accelerated the inactivation kinetics of Na$_V$1.4 to 5 ± 0.9 ms and 8 ± 1.0 ms, respectively, while not being significantly different from each other (p=0.08). Analysis includes data from two oocyte batches. Values are presented as mean ± s.e.m.

The following figure supplements are available for figure 3:

**Figure supplement 1.** Iodide blocks ClC-0 E166Phe currents, indicating that they are indeed ClC-0 mediated.

**Figure supplement 2.** Nonsense suppression efficiency at position E166 in ClC-0.

cell-free protein synthesis, Cy3 and Cy5-ncAAs could be encoded at E166*TAG*, resulting in functional channels with robust expression levels, constitutively open gating and chloride selectivity, similar in each parameter to Phe incorporation (*Figure 3A–C*). To further test the ncAA size limits of the eukaryotic ribosome, a 'self-healing' Cy3 variant LD550 (*Zheng et al., 2014*) was encoded at the E166 position in ClC-0. The macroscopic currents elicited by the fluorophore-incorporated ClC-0 channels were all well above the background currents (Cy3: 8.6-fold; Cy5: 5.8-fold; LD550: 4.8-fold; p<0.001 for all *Figure 3B*), thus demonstrating significant suppression efficiencies of the encoded Cy-ncAA-tRNA. The importance of optimizing the tRNA ligation reaction is highlighted when comparing the expression of channels rescued by tRNA produced at 37°C versus 4°C where the former condition yields less efficient channel rescue levels (*Figure 3B*). Further, the amount of tRNA used was also optimized for suppression at ClC-0 E166*TAG* (*Figure 3—figure supplement 2*). Of note, due to the central location of the E166 encoding site within the ClC-0 reading frame, truncated protein transcripts that lack an encoded Cy-ncAA would be non-functional and biologically inert. Thus, the whole-cell ionic currents are produced by a substantial population (~750,000 channels per μA [*Ludewig et al., 1996*]) of functionally expressed, full-length chloride channels at the plasma membrane that contain the encoded Cy-ncAA.

To further validate the approach, Cy3-AF was encoded in the eukaryotic sodium channel (Na$_V$) complex. Na$_V$s contribute to the upstroke of the action potential in the excitable tissues of nerve and muscle (*Ahern et al., 2016*), are widely implicated in inherited and acquired human diseases and are high value therapeutic targets (*Ahuja et al., 2015*). The sodium ion selective pore forming α-subunit displays modulated expression and gating by the single pass transmembrane β subunit (*O'Malley and Isom, 2015*). Specifically, in the context of the *Xenopus* oocyte, co-expression of Na$_V$1.4 and β1 produces currents with accelerated inactivation compared to Na$_V$1.4 expressed alone (*Makita et al., 1996*). The data show that encoding Cy3 at position E27 in the extracellular domain of β1 resulted in acceleration of Na$_V$1.4 inactivation kinetics comparable to wild-type β1 with minimal read-through, consistent with the functional expression of the full-length auxiliary β1 subunit (*Figure 3D and E*). Thus, in tandem, the electrophysiological data demonstrate the successful site-specific incorporation of Cy-ncAAs into two different membrane proteins – ClC-0 and Na$_V$ β1 subunit.

To confirm plasma membrane expression of ClC-0 channels containing encoded Cy-ncAAs, oocytes were examined using a single fluorescent molecule TIRF microscope. Under TIRF illumination multiple fluorescent spots could be detected in the region of the oocyte plasma membrane in contact with the coverslip glass (*Figure 4A*). Oocytes were coinjected with ClC-0 E166*TAG* cRNA + Cy3-AF-tRNA + Cy5-AF-tRNA or for the mock condition with ClC-0 wild-type cRNA + Cy3-AF-tRNA + Cy5-AF-tRNA to estimate non-specific background fluorescence. The analysis yielded three significant observations that demonstrate the functional encoding of the Cy-ncAAs into ClC-0. First, overall Cy3 and Cy5 spot densities were higher for the encoded (Cy-ncAA-tRNA plus ClC-0 E166*TAG* cRNA) versus 'mock' conditions (Cy-ncAA-tRNA plus ClC-0 wild-type cRNA; *Figure 4A*

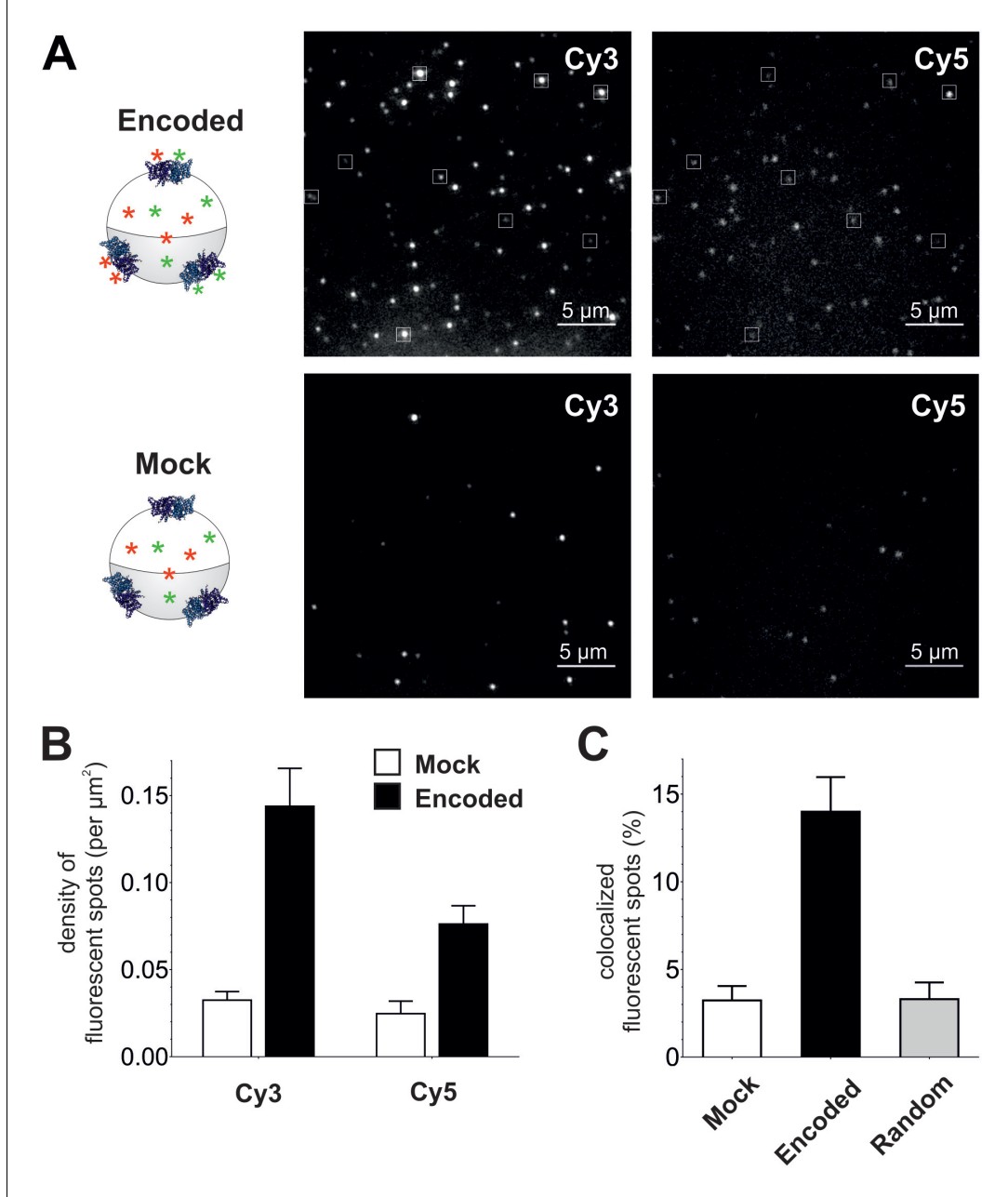

**Figure 4.** Single-molecule TIRF microscopy imaging of cellular encoded Cy3 and Cy5 fluorophores in ClC-0, a dimeric plasma membrane chloride channel. (**A**) TIRF images show distinct fluorescent spots for Cy3 and Cy5 encoded into ClC-0 at position E166 (Encoded (top), Mock (bottom)). A subpopulation of Cy3 and Cy5 spots colocalized (highlighted in white boxes). The encoded and mock injected images were grayscaled to identical levels to enable viewing of both bright and dim spots. (**B**) Number density of fluorescent spots per area ($\mu m^2$) of plasma membrane was significantly different between Encoded (black) and Mock (white). p values see main text. Total numbers of counted spots included in this analysis are: N = 4406 for encoded Cy3, N = 2459 for encoded Cy5, N = 1154 for mock Cy3, N = 717 for mock Cy5; they result from ≥5 oocytes (≥3 batches) per condition (**C**) Same images as in (**B**) were examined for colocalization between Cy3 and Cy5. In Encoded, 14 ± 2% of the Cy5 signal merged with Cy3. Random colocalization in those images was estimated to 3 ± 1% and colocalization in Mock was not significantly different from that (p=0.96). All values are mean ± s.e.m.

*and B*). Specifically, the number density for the encoded Cy3 was $0.14 \pm 0.02$ spots per $\mu m^2$ and $0.08 \pm 0.01$ for the encoded Cy5, while in mock $0.03 \pm 0.01$ spots per $\mu m^2$ were detected for Cy3 (p=0.001) and $0.02 \pm 0.01$ for Cy5 (p=0.004; *Figure 4B*). The fluorescent spots observed in the mock condition may be ascribed to non-encoded Cy-ncAA-tRNA or hydrolyzed Cy-ncAA that may have escaped the cytosol during peeling the vitelline membrane (a prerequisite to TIRF imaging) or that are trapped within the small cytosolic volume of the oocyte microvilli (*Sonnleitner et al., 2002*). The likelihood that those spots contain CyX-ncAAs encoded within the truncation codons of endogenous plasma membrane proteins that use TAG as a native stop codon is very low because their expression level well below of an overexpressed protein. Further, analysis of EST databases for *Xenopus laevis* oocytes stage V and VI shows that the list of possible candidates is limited (*Supplementary file 1*). However, most importantly, read-through of a stop codon of an endogenous protein would likely lead to translation of the 3' UTR and ubiquitin-mediated protein degradation (*Bengtson and Joazeiro, 2010*). The second observation supporting specific encoding of CyX-ncAAs into ClC-0 is the significant colocalization between Cy3 and Cy5 fluorescent spots in oocytes co-injected with Cy3, Cy5 and ClC-0 E166*TAG* cRNA. Of all imaged spots, $14 \pm 2.2\%$ of those with Cy5 fluorescence co-localized with Cy3 spots in the encoded reaction, compared to only $3 \pm 1.0\%$ for randomly co-localized Cy3/Cy5 spots, an outcome similar to mock conditions with $3 \pm 0.8\%$ (*Figure 4A and C*). Thus, in dual Cy3- and Cy5-ncAA encoding conditions, multiple fluorophores appear within single spots suggesting either the encoding of Cy3- and Cy5-ncAA in single or multiple ClC-0 dimers within a distinct fluorescent spot. Lastly, a photo-bleaching analysis of fluorescent spots in encoded and mock condition was performed (*Figure 5*, *Figure 5—figure supplement 1*). Multiple examples of step-wise bleaching events of the encoded condition are presented in *Figure 5A and B*. The data reveal that while the mock bleaching profile was strongly dominated by single step events, the encoded reaction showed a clear shift towards multiple steps (2+), as shown as absolute values in a histogram (*Figure 5C*) and in relative values (*Figure 5—figure supplement 1*). Further, under encoding conditions some spots showed upwards of 4- and 5-step bleaching events, an observation that was unmatched in the mock condition. Thus, the photo-bleaching data further demonstrate encoding of the Cy-ncAA, likely within single and multiple ClC-0 dimers, in a manner that is consistent with expression beyond the single-molecule level or clustering of rescued full-length channels. Indeed, clustering of ClC-0 wild-type channel has been observed previously in excised patch experiments (*Bauer et al., 1991*).

None the less, the photo-bleaching data, while strongly supportive of the encoding of the Cy-ncAA, also indicate that there remain a number of issues that must be overcome before using this method generally for stoichiometry measurements. First, the basis for the observation of spots that bleach with >2 steps merits further investigation. This could arise, as noted, from over-expression of ClC-0 in the membrane, a possibility that could be tested by titrating down to single-molecule levels by reducing the amount of tRNA injected or reducing expression time. Another possibility is that there are truly higher-order assemblies of ClC-0 in the membrane, as has been observed before and described as clustering (*Bauer et al., 1991*). In this case, ClC-0 may not be a good system for studying stoichiometry. However, ClC-0 was chosen as a model membrane protein with proven expression in the oocyte, with the added benefit that its permeation pathway has been well studied with the identification of residues such as E166 that are known to be tolerant of a variety of side-chains. A second issue is that the labeling yield is too low to allow for reliable stoichiometry analysis. The data from HPLC analysis of the tRNA suggest that this decrease in fluorescence yield occurs after injection into the oocyte, but it could be occurring after the dye is in the extracellular oxygen rich environment of the solution or during preparation for imaging. During incorporation, the Cy-ncAA may be exposed to reductive conditions in the cell that lead to formation of dark state adducts (*Dempsey et al., 2009*). Furthermore, imaging was performed in the absence of typical oxygen scavenger systems, as this has been shown to lead to improved photo-bleaching traces (*Chadda et al., 2016*) but could reduce the overall fluorescent yield. In this regard, future encoding experiments performed in the presence of oxygen quenchers or more stable fluorophores (*Zheng et al., 2014*) may be required for specific applications of the approach. Overall, in its current iteration, the method will require more development in order to be useful for stoichiometry measurements of protein complexes mainly because of uncertainty in labelling efficiency. None the less, once these issues have been overcome, the use of such probes will be a significant advance for stoichiometry studies

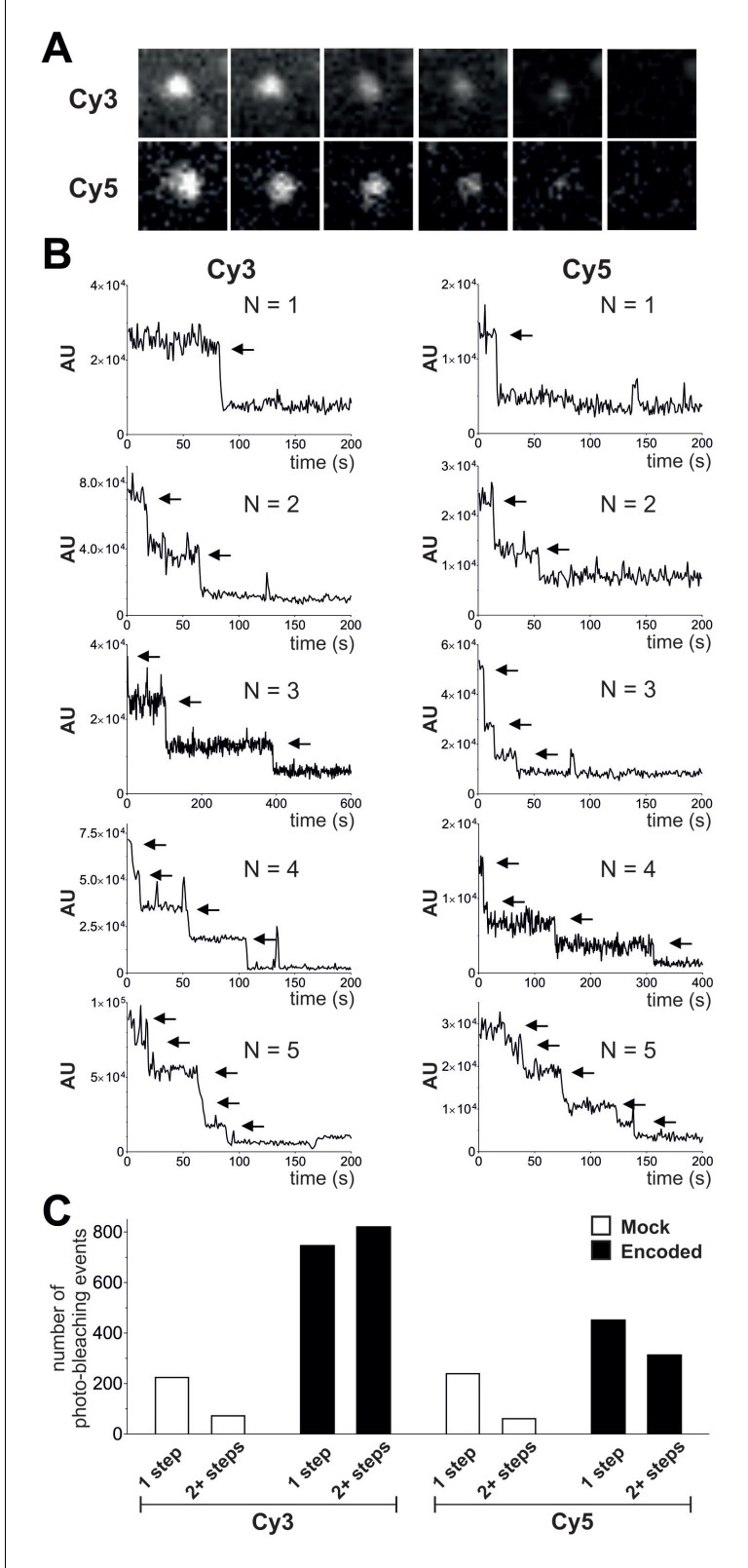

**Figure 5.** Photo-bleaching profiles for Cy3 and Cy5 spots in Encoded and Mock conditions. (**A**) and (**B**) Photo-bleaching events of Cy3 and Cy5 fluorescent spots in the Encoded sample. (**A**) Example of a 5-step photo-bleaching event of a fluorescent spot showing Cy3 (top) or Cy5 fluorescence (bottom), respectively, in an oocyte co-injected with ClC-0 E166*TAG* cRNA as well as Cy3- and Cy5-tRNA. A single representative frame was selected

*Figure 5 continued on next page*

*Figure 5 continued*

from the video sequence showing step-wise reduction in spot-intensity. Presented area size is 20 pixels x 20 pixels. (B) Examples of fluorescence traces (in arbitrary units, AU) as a function of time show step-wise photo-bleaching for Cy3 (left) and Cy5 (right), respectively, until complete photo-destruction of the fluorophore was achieved. Arrows indicate observed steps during the bleaching process; N is equal to the total number of identified steps. (C) Photo-bleaching histogram for Cy3 and Cy5 spots in Encoded and Mock reveal a distribution distribution of steps: while in Mock 1-step events strongly dominated, in Encoded the distribution was clearly shifted towards 2+ steps. Values represent absolute numbers for the identified bleaching step event analyzed from an equal number of frames for each condition (five oocytes, three batches). In total, following numbers were included: N = 1564 for encoded Cy3, N = 762 for encoded Cy5, N = 296 for mock Cy3, N = 300 for mock Cy5.

The following figure supplement is available for figure 5:

**Figure supplement 1.** Photo-bleaching profiles for Cy3 and Cy5 spots in Encoded and Mock.

---

over current approaches that rely of chemical labeling of proteins with Cy-dyes or encoded probes, such as GFP.

However, examining dynamic single-molecule FRET is possible, in principle, provided only spots containing both Cy3 and Cy5 fluorescence signals are examined. Yet, due to low degree of colocalization this may be impractical depending on the target protein. The method could also be applied to extract time-regimen information from single-molecule experiments. In fact, estimating lateral diffusion in cellular plasma membrane by single molecule tracking necessitates sparse labelling. In the past, these studies have led to discovery of dynamic nano lipid-raft formation (*Suzuki et al., 2012*) or elucidation of organization of focal adhesions (*Kusumi et al., 2014*). It is also useful for other single-molecule studies such as ligand binding (e.g. kinetics of toxin binding, or other high-affinity ligands) or a single-molecule version of voltage fluorometry. Thus, while additional effort is needed to maximize the experimental possibilities for encoding Cy-ncAA, there are a number of applications at this stage. Further, we anticipate that ultimately it will be applicable to mammalian cell lines once approaches for tRNA delivery have been established.

The incorporation of noncanonical amino acids via nonsense suppression with in vitro acylated tRNAs results in significantly reduced yields compared to wild-type expression (*Leisle et al., 2015*; *Dougherty and Van Arnam, 2014*). However, the eventual goal of encoding Cy-ncAA is for single-molecule studies, thus, decreased yield is not necessarily a barrier to the application of this approach. In the case of ClC-0, for example, the injection of 100-fold less of the wild-type cRNA was needed to match the lower current amplitudes observed in the 'rescued' ClC-0 channels (*Figure 3— figure supplement 2*). There are multiple factors that impact the overall efficiency of suppression. For one, competition between the supplied suppressor tRNA with the endogenous translation terminator, release factor 1 (RF-1), for TAG codons. Second, the intrinsic suppression competency of in vitro transcribed and folded tRNA compared to that of native tRNA is not known. Consequently, the in vitro synthesized, orthogonal tRNA might display a lower translation efficiency compared to native tRNAs. Third, amino acid hydrolysis from the tRNA is pH and temperature dependent and might occur during incubation of the injected cells prior to forming a complex with the endogenous elongation factors (*Peacock et al., 2014*; *Stepanov and Nyborg, 2002*).

Taken together, the data demonstrate that Cy-ncAA-based optical probes were genetically incorporated into membrane proteins (providing two examples: a chloride ion channel and a voltage-gated sodium channel auxiliary subunit), and that these proteins were functional and trafficked to the cell surface. TIRF microscopy was used to obtain single-molecule imaging and photo-bleaching data from cells expressing membrane proteins containing Cy3 and Cy5, further confirming that encoding was successful. Thus, the study provides the first description of the genetic encoding of organic cyanine dyes by a eukaryotic ribosome in a cell-free translation system as well as in live cells. The observation that Cy-ncAAs can be encoded into a nascent transcript by the eukaryotic ribosome reveals that the ribosome is surprisingly plastic, and thus opens the door to study proteins by single-molecule approaches in cellular environments. The current fluorescent yields make studies such as ligand binding, or voltage-dependent fluorescence experiments possible. Once fluorescent yields are improved, the approach has applications for the future study of multisubunit stoichiometry

through photo-bleaching of encoded Cy-ncAAs, or for obtaining real-time insights into conformational dynamics by single-molecule FRET studies in live cells.

## Materials and methods

### Synthesis of CyX non-canonical amino acids (CyX-ncAA)

The ncAAs Cy3-, Cy5- and LD550-para-amino-L-phenylalanine were synthesized conjugated to the dinucleotide phospho-desoxy-cytosine phospho-adenosine (pdCpA or short pCA).

### General Information

All solvents and reagents were supplied by Sigma Aldrich (St. Louis, MO) and were used as-is unless explicitly stated and pdCpA (pCA) was obtained through GE Healthcare/Dharmacon (LaFayette, CO). Cy3 and Cy5 NHS esters were purchased from Click Chemistry Tools (Scottsdale, AZ) as tri-sulfates. Of note, one of the sulfate groups on the fluorophores was a propyl-sulfate. This propyl-sulfate group has been replaced by a methyl group by the vendor and the di-sulfate CyX NHS ester products are now sold under the same catalogue number. LD550 NHS ester was supplied by Lumidyne Technologies (New York, NY). Dry nitrogen was supplied by Praxair (Danbury, CT) and passed through two moisture scrubbing columns of dry calcium sulfate (Drierite) prior to use. HPLC analyses were performed on a Waters (Milford, MA) 1525 Binary HPLC pump equipped with a Waters (Milford, MA) 2998 Photodiode Array Detector, employing Sunfire C18 analytical (3.5 μm, 4.6 mm x 150 mm, 0.8 ml/min) or preparative (5.0 μm, 19 mm x 150 mm, 10 ml/min) columns and Empower software, buffers were drawn in linear gradients from 100% A (50 mM ammonium acetate) to 100% B (acetonitrile) over 30 min. UV-visible spectra for concentration determinations were recorded on a Thermo Scientific (Grand Island, NY) Nanodrop 2000C spectrophotometer. Mass spectra were recorded on a Waters QToF Premier Quadrupole instrument, in both positive and negative modes.

### AF-pCA

Slight modifications from the published procedure (*Hohsaka et al., 1999*) were employed. The N-Boc protected cyanomethyl ester of AF (75 mg, 0.23 mmol) and pCA (30 mg, 0.045 mmol) were dissolved in dry dimethylformamide (0.5 ml) in a round-bottom flask (5 ml) and tetrabutylammonium acetate (30 mg, 0.1 mmol) was added. The solution was stoppered and stirred at room temperature and monitored by HPLC over several hours (2 μl aliquots were removed and diluted into 100 μl of a 4:1 mixture of A:B buffer prior to injection) with detection at 240 nm. Unreacted pCA elutes at 10.1 min, Boc-AF-pCA elutes at 13.7 min and Boc-AF-CN ester elutes at 21.8 min. The reaction was judged complete after consumption of free pCA, after ~4 hr. At this scale the reaction was divided equally into 8 × 1.5 ml tubes and ~1 ml of ice-cold ether was introduced to each, resulting in a cloudy precipitate, which was centrifuged down to a pellet (10,000 rpm, 1 min). The eight pellets were then each resuspended in 50 μl acetonitrile and precipitated again with 1 ml ether. Following a third round of precipitation the eight tubes were combined into one and precipitated a fourth time, then dried in a gentle stream of dry nitrogen gas after centrifugation. Ice-cold trifluoroacetic acid (~300 μl) was added and the pellet was dissolved via agitation and pipetting and placed on an ice bath for 30 min to remove the Boc group. Excess trifluoroacetic acid was removed in a gentle stream of dry nitrogen gas until a sticky oil remained, and ice-cold ether was added to precipitate the TFA salt of AF-pCA, which was washed twice more with ether and very carefully dried in a gentle stream of dry nitrogen gas. The resulting granular powder was stored as a salt at −20°C. Mass spectrometry confirmed the identity of the molecule (LRMS (m/z): [M-H]⁻ calculated for $C_{28}H_{35}N_{10}O_{14}P_2$, 797.18; found, 797.2). All results are shown in *Figure 1—figure supplement 1*.

### CyX-AF-pCA

Slight modifications from the published procedure were employed (*Kajihara et al., 2006*; *Watanabe et al., 2007*). AF-pCA TFA salt (1 mg) was dissolved in 1 M aqueous pyridine HCl (50 μl, pH 5.0) and added to CyX NHS ester (0.5 mg, ~0.5 μmol) in dimethylsulfoxide (50 μl) in a 1.5 ml tube, and an extra 50 μl of pyridine HCl was added before heating the solution to 37°C and was shielded from light with foil. The reaction was monitored by HPLC over several hours (1 μl aliquots

were removed and diluted into 100 µl of A buffer prior to injection) with detection at 550 nm (Cy3 and LD550) or 650 nm (Cy5). Product peaks are identified by retention time shift and UV absorptions characteristic of both the Cy chromophore (550/650 nm) and the cytosine/adenosine bases in pCA (260 nm) (*Figure 1—figure supplements 2A–4A*). The reaction was terminated after several hours by dissolving in buffer A (4 ml) and immediate injection into the HPLC. Fractions containing product were frozen on dry ice and lyophilized overnight while shielded from light. The resulting residue was carefully dissolved in ~50 µl dry DMSO and checked for product integrity via HPLC and stock concentration was approximated on a NanoDrop UV-vis spectrometer using the known molecular extinction coefficients for the Cy dyes. Each CyX-AF-pCA isolated in this manner had its concentration adjusted to 3 mM with DMSO and the stock was stored in aliquots at −28°C. Mass spectrometry confirmed the identity of the molecules (LRMS (m/z): Cy3-AF-pCA [M-H]$^-$ calcd. for $C_{60}H_{73}N_{12}O_{24}P_2S_3$, 1503.35; found, 1503.3, [M-2H]$^{-2}$ calcd., 751.17; found 751.2. Cy5-AF-pCA [M-H]$^-$ calcd. for $C_{62}H_{75}N_{12}O_{24}P_2S_3$, 1529.36; found, 1529.4, [M-2H]$^{-2}$ calcd., 764.18; found 764.2; *Figure 1—figure supplements 2C–3C*). Of note, for the LD550 NHS ester no structure was provided by Lumidyne Technologies which prevented a mass spectrometry analysis of the final product LD550-AF-pCA.

## tRNA transcription and misacylation

For nonsense suppression during cell-free translation, pyrrolysine tRNA (PylT) was used. A modified (G73) version of *Tetrahymena thermophila* tRNA, THG73 was employed for nonsense suppression in *Xenopus leavis* oocytes. To yield a full-length (acylated) tRNA, first a construct lacking the last two nucleotides was transcribed in vitro using CellScript T7-Scribe Standard RNA IVT Kit (CELLSCRIPT, Madison, WI). The DNA templates (coding for the tRNA preceded by a T7 promoter) were ordered as PAGE-purified primers from Integrated DNA Technologies (Coralville, IA). THG73 forward: ATTCGTAATACGACTCACTATAGGTTCTATAGTATAGCGGTTAGTACTGGGGACTCTAAA TCCCTTGACCTGGGTTCGAATCCCAGTAGGACCGC; THG73 reverse: GCGGTCCTACTGGGA TTCGAACCCAGGTCAAGGGATTTAGAGTCCCCAGTACTAACCGCTATACTATAGAACCTATAG TGAGTCGTATTACGAAT; PylT forward: ATTCGTAATACGACTCACTATAGGAAACCTGATCATG TAGATCGAACGGACTCTAAATCCGTTCAGCCGGGTTAGATTCCCGGGGTTTCCGC; PylT reverse: GCGGAAACCCCGGGAATCTAACCCGGCTGAACGGATTTAGAGTCCGTTCGATCTACATGA TCAGGTTTCCTATAGTGAGTCGTATTACGAAT. 20 µg of annealed oligonucleotides were used as a template. The total reaction volume was adjusted to 100 µl and the kit reagents were added in the following amounts: 10 µl of 10X T7-Scribe transcription buffer, 7.5 µl of each nucleotide (100 mM stocks), 10 µl of 100 mM Dithiothreitol, 2.5 µl ScriptGuard RNase Inhibitor, 10 µl T7-Scribe enzyme solution. After the reaction was incubated for 4–5 hr at 37°C, the DNA template was digested with 5 µl DNase (1 U/µl) provided with the kit for 30–60 min. The tRNA was extracted from the reaction with acidic phenol chlorohorm (5:1, pH 4.5) and precipitated with ethanol. The precipitates tRNA was pelleted, washed, dried and resuspended in 100 µl DEPC-treated water and further purified with Chroma Spin-30 columns (Clontech, Mountainview, CA). The procedure yielded roughly 100 µl of ~5 µg/µl tRNA that was stored in aliquots at −80°C. Prior to the ligation reaction of the 73-mer THG73 to the ncAA-pCA conjugate (or just the pCA to yield a complete, nonacylated tRNA), the tRNA was folded in 10 mM HEPES (pH 7.4) by heating at 94°C for 3 min and subsequent gradual cool-down to ~10°C. 25 µg of folded tRNA in 30 µl of 10 mM HEPES (pH 7.4) were mixed with 28 µl of DEPC-treated water, 8 µl of 3 mM ncAA-pCA (or pCA) dimethylsulphoxide stock, 8 µl of 10X T4 RNA Ligase one buffer, 1 µl 10 mM ATP and 5 µl of T4 RNA Ligase 1 (New England Biolabs, Ipswich, MA) then incubated at 4°C for 2 hr (for experiments to optimize the acylation efficiency the following conditions were also applied: 37°C, 40 min; 4°C, 5 hr; 4°C, 11 hr). The (misacylated) tRNA was extracted from the samples with acidic phenol chlorohorm (5:1, pH 4.5) and precipitated with ethanol. The precipitated tRNA pellets were washed, dried in a Speedvac and stored at −80°C.

The ligation efficiencies of the CyX-tRNAs were determined by absorbance measurements at 260 nm (RNA) and 550 nm (Cy3/LD550) and 650 nm (Cy5), respectively, in serial dilutions and duplicates for each resuspended tRNA pellet. The calculations were performed according to the Lambert-Beer law using the following molar extinction coefficients: 696,100 M$^{-1}$ cm$^{-1}$ for THG73 tRNA (http://www.idtdna.com/calc/analyzer), 150,000 M$^{-1}$ cm$^{-1}$ for Cy3 and 250,000 M$^{-1}$ cm$^{-1}$ for Cy5; and the

following equation: ligation efficiency (%) = $[A_{CyX}*\varepsilon_{RNA}/(A_{RNA}*\varepsilon_{CyX})]*100\%$. In total 3–5 acylated tRNA pellets were used independently per condition for each fluorophore.

In the attempt to purify acylated from non-acylated tRNA after the ligation reaction the following HPLC purification protocol was employed. CyX-acylated tRNA pellets were dissolved in 100 µL of buffer A (20 mM ammonium acetate, pH 5.0 containing 10 mM magnesium acetate and 400 mM sodium chloride), loaded onto a POROS R2/10 column (2.1 mm x 100 mm, Applied Biosystems, Carlsbad, CA) and separated utilizing a series of linear gradients with increasing amounts of buffer B (70% buffer A, 30% ethanol). A Waters 2998 photodiode array detector enabled detection and isolation of peaks corresponding to both free and acylated tRNA (*Figure 2—figure supplement 1*). The efficiency of the CyX-tRNA ligation was estimated based on the area under the HPLC peaks at 260 nm. Due to extremely high loss of material during recovery from purification, the HPLC-purified tRNA was not utilized in experiments presented, thus the data serves to confirm the determination of the ligation efficiencies by absorbance measurements (see above).

## Nonsense suppression in eukaryotic cell-free expression system

All cell-free translation reactions were performed using the nuclease-treated Rabbit Reticulocyte Lysate System (Promega, Madison, WI). The cRNAs for NanoLuc Amber (NanoLuc luciferase with an inserted amber stop codon between G159 and V160) was generated from a linearized pcDNA3.1 plasmid containing the gene of interest using the mMessage mMachine T7 Ultra Kit (Thermo Fisher Scientific, Grand Island, NY). Subsequently, the cRNA was purified with RNeasy Mini Kit (Quiagen, Hilden, Germany), the concentration was determined by absorbance measurements at 260 nm and the quality was confirmed on an RNase-free 1% agarose gel. The tRNA transcription and misacylation is described above. For nonsense suppression of the luciferase construct PylT tRNA was used. The cell-free translation reactions were assembled according to the user's manual: 35 µl lysate, 0.5 µl amino acid mixture minus Leu (1 mM), 0.5 µl amino acid mixture minus Met (1 mM), 1 µl RNasin Ribonuclease inhibitor (40 U/µl), 1 µl cRNA (2 µg/µl), 1.5 µl nonacylated or acylated tRNA (10 µgl/µl), 10.5 µl DEPC-treated water. The reaction took place at 22°C for 90 min.

To quantify the luciferase activity the reactions were diluted 1:10 in PBS (pH 7.4) and mixed 1:1 with the reagent from Nano-Glo Luciferase Assay System (Promega, Madison, WI). The measurements were conducted in triplicates on the plate reader Spectramax i3 (Molecular Devices, Sunnyvale, CA) at room temperature and 1 s integration time. All experiments were performed three times.

## Nonsense suppression in *Xenopus laevis* oocytes

The technique was applied as previously described (*Pless and Ahern, 2013*; *Dougherty and Van Arnam, 2014*; *Leisle et al., 2015*) with modifications. The tRNA transcription and misacylation is described above. Regarding the in vitro cRNA transcription, the cRNA for ClC-0 and ClC-0 E166TAG was transcribed from a pTLN vector using the mMessage mMachine SP6 Kit (Thermo Fisher Scientific, Grand Island, NY). Rat Na$_V$β1 and β1 E27*TAG* were transcribed from a pcDNA3.1 plasmid using the mMessage mMachine T7 Kit (Thermo Fisher Scientific, Grand Island, NY) and Na$_V$1.4 from a pBSTA vector with the mMessage mMachine T7 Ultra Kit (Thermo Fisher Scientific, Grand Island, NY). All reactions were set up according to the user manuals. Purification of the cRNA from the transcription reaction was conducted on columns from the RNeasy Mini Kit (Quiagen, Hilden, Germany). Concentration was determined by absorbance measurements at 260 nm and quality was confirmed on a 1% agarose gel (RNase-free).

### Protein expression in Xenopus laevis oocytes

*Xenopus laevis* oocytes (stage V and VI) were purchased from Ecocyte (Austin, TX). Prior to injection each tRNA pellet was resuspended in 1.5 µl of 3 mM sodium acetate (to avoid hydrolysis of the amino acid from the tRNA) and span down at 21,000 xg, 4°C for 25 min. For two electrode voltage clamp experiments the following amounts were injected: in the case of chloride channels, 7.5 ng of ClC-0 E166*TAG* cRNA with 190–250 ng of nonacylated or misacylated tRNA (unless specified otherwise), and, in the case of sodium channels, 1 ng Na$_V$1.4 cRNA or 1 ng Na$_V$1.4 cRNA with 2 ng of rat β1 cRNA or 1 ng Na$_V$1.4 cRNA with 20 ng of rat β1 E27*TAG* cRNA and 150–200 ng of nonacylated or misacylated tRNA. After injection the oocytes were kept in OR-3 (50% Leibovitz's medium, 250

mg/l gentamycin, 1 mM L-glutamine, 10 mM HEPES pH = 7.6) at 18°C for ~1 d till measurable currents were detected. For the TIRF microscopy experiments, oocytes were pre-incubated in hypertonic OR-2 and subsequently recovered in isotonic OR-2 before injection (details see *Sample preparation* in *Single-molecule imaging via Total Internal Reflection Fluorescence (TIRF) microscopy*). Here, the following amounts were injected: for the encoded reaction, 7.5 ng of ClC-0 E166*TAG* cRNA with 190–250 ng of nonacylated or misacylated tRNA (Cy3- or Cy5-ncAA misacylated tRNAs in a 1:1 ratio); for the mock control, 0.5 ng ClC-0 mRNA with 190–250 ng of misacylated tRNA. Protein expression was confirmed prior to single-molecule imaging via TEVC. Wildtype cRNAs were injected in 10-fold reduced amounts to yield comparable current amplitudes. The total injection volume was kept constant at 50 nl for all conditions.

## Two electrode voltage clamp (TEVC) recordings

TEVC was performed as described before extensively (*Pless et al., 2011*; *Pusch et al., 1995*). In brief, voltage-clamped chloride currents were recorded in ND96 solution (in mM: 96 NaCl, 2 KCl, 1.8 CaCl2, 1 MgCl2, 5 HEPES, pH 7.5) and sodium currents in standard Ringer (in mM: 116 NaCl, 2 KCl, 1 MgCl2, 0.5 CaCl2, 5 HEPES, pH 7.4) using an OC-725C voltage clamp amplifier (Warner Instruments, Hamden, CT). For Iodide exchange experiments of ClC-0 currents, NaCl was substituted by NaI equimolar. Glass microelectrodes backfilled with 3 M KCl had resistances of 0.5–3 MΩ. Data were filtered at 1 kHz and digitized at 10 kHz using a Digidata 1322A (Molecular Devices, Sunnyvale, CA) controlled by the pClamp 9.2 software (Molecular Devices, Sunnyvale, CA). Chloride currents were elicited by +20 mV voltage steps from −120 mV to +80 mV preceded by a +80 mV prepulse from a holding potential of −30 mV. Sodium currents were elicited by a 50 ms test pulse to −20 mV from a holding potential of −120 mV. Clampfit 9.2 software was used for current analysis. Numbers of oocytes and batches used, details on current analysis and statistical significance of effects are indicated in the appropriate figure legends or the main text.

## Single-molecule imaging via total internal reflection fluorescence (TIRF) microscopy

### Sample preparation

TIRF microscopy generates an evanescent wave that excites fluorophores in a restricted region of the specimen immediately adjacent to the glass-water interface. As the evanescent wave decays exponentially, it is effective only in the 100–250 nm range enabling the selective visualization of the plasma-membrane and cell cortex, thus minimizing the background fluorescence from the cytosol. To expose the oocyte plasma membrane, the ~1 μm thick vitelline membrane (*Sonnleitner et al., 2002*) was removed mechanically with forceps under a dissecting microscope after incubating the oocytes in hypertonic OR-2 solution (in mM: 82.5 NaCl, 2.5 KCl, 1 MgCl$_2$, 5 HEPES pH 7.5) for max. 5 min (osmolarity of the hypertonic solution was adjusted with sucrose to 500 mOsm/l). Incubation in hypertonic ringer helps to separate the vitelline membrane from the plasma membrane as the oocytes shrink in size. In order to minimize complicating effects of micro-ruptures of the plasma-membrane (not visible under the dissecting microscope) which may result in the leakage of fluorophores from the cytosol, two additional preparatory steps were performed. First, before injection of the oocytes the connections between the vitelline membrane and the plasma-membrane were weakened by incubation in hypertonic OR-2 for max. 5 min (as inspired by [*Choe and Sackin, 1997*]). Subsequent recovery in isotonic OR-2 was allowed before injection. Second, after peeling the vitelline membrane on the day of imaging, the oocytes were transferred from the hypertonic solution into the slightly hypertonic OR-2 with 250 mOsm/l and incubated on an orbital shaker at ~30 rpm for at least 20 min. The oocytes were incubated singly in separate wells of a non-treated 6-well plate (with 4 ml solution per well). Only the oocytes that did not burst or loose their shape during that procedure were subjected to single-molecule imaging. They were washed twice in 250 mOsm/l OR-2 and transferred onto a treated glass coverslip (oriented with the animal pole towards the glass) where they were incubated for ~15 min prior to imaging to maximize their adherence.

### Glass slide preparation

Glass slides (Gold-Seal 24 × 60 mm, no.1.5 thickness; Thermo Fisher Scientific, Grand Island, NY) were cleaned by the following procedure: 30 min sonication in 0.1% Micro-90 detergent solution

(Cole-Parmer, Vernon Hills , IL), 5 vol washes with MilliQ-grade ddH$_2$O, 30 min sonication in absolute ethanol, 5 vol washes with MilliQ-grade ddH$_2$O, then 5 min sonication in 0.2 M KOH. After five final washes in MilliQ-grade ddH2O, the glass slides were stored in a clean air laminar flow hood at RT for 1–2 weeks. On the day of imaging, rectangular chambers were drawn with silicon vacuum grease on the cleaned glass slide. To maximize the adherence of the oocyte, the glass bottom of the chamber was coated with 0.1 mg/ml poly-L-lysine solution and letting it evaporate in the clean hood.

## TIRF microscopy

Images were collected on a custom built multi-wavelength single-molecule TIRF microscope following the CoSMoS design (*Larson et al., 2014*; *Friedman et al., 2006*), equipped with three excitation lasers: OBIS 488 and 637 nm, and Sapphire 532 nm (Coherent Inc., Santa Clara, CA) that are directed into a 60 × 1.49 NA objective (Olympus Corp., Tokyo, Japan) using a Micro-Mirror TIRF platform (Mad City Labs Inc., Madison, WI). The emitted fluorescence was collected using a broadband mirror and focused onto an iXon3 888 EM/CCD camera (Andor Inc., South Windsor, CT). The illuminated area on the coverslip was ~2500 µm (*Zachariassen et al., 2016*). A region of the oocyte was selected to focus the sample and then adjust the TIRF angle. Then images for analysis were collected on adjacent fields which were unexposed to laser illumination. The images were acquired in a sequential manner. First, oocytes were excited with 637 nm laser (OBIS; Coherent Inc., Santa Clara, CA) and emitted light was collected using a notched Cy5 emission filter (ZET635nf; Chroma, Bellows Falls, VT). After all the Cy5 spots were photobleached excitation was switched to 532 nm (Sapphire; Coherent Inc., Santa Clara, CA) to excite and photobleach Cy3 spots. Cy3 emission was collected using a bandpass filter (ET585/40m; Chroma, Bellows Falls, VT). Image sequences were acquired at ~1 frame per second with EM gain set to 300, and laser intensity at 1.2 mW for Cy3 and 250 µW for Cy5 line to obtain long photo-bleaching traces while maintaining good signal to noise ratios. We consistently observed higher background in encoded-Cy5 images possibly related to greater penetration depth of evanescent field at longer wavelengths.

## Photo-bleaching analysis

Images were acquired as stacked tiff files. Background subtraction was performed in ImageJ (*Schneider et al., 2012*) using a rolling ball radius of 50 pixels (*Sternberg, 1983*). The resultant images were analyzed using a Matlab-based CoSMoS analysis software (*Friedman and Gelles, 2015*). There, 7 pixel x seven pixel areas of interest (AOIs) were auto-detected around each fluorescent spot based on intensity threshold (Cy3 fluorescence was 150–190 grayscale units and Cy5 130–150 gray scale units) and area, in the first frame of each image sequence. Integrated fluorescence intensity from the AOIs was plotted as a function of time for each AOI allowing for analysis of step-like photo-bleaching behavior (*Figure 5B*). The traces were classified manually according to the number of observed photo-bleaching steps before complete photo-destruction of the fluorophore occurred. The total occurrence of each step event was calculated for each movie and then pooled for presentation in text and figures (same number of movies was used for each condition). Relative occurrence of each step event was calculated for each movie, then averaged per oocyte and subsequently pooled to calculate the mean ± s.e.m. for presentation in text and figures.

## Co-localization analysis

A crude Cy3-Cy5 colocalization analysis was performed. Images were background subtracted and Cy5 AOIs were identified as described above. These AOIS were collectively mapped onto the Cy3 field and then manually checked for the presence of a Cy3 spot. Observation of constellations of co-localized spots allowed us to confirm that the mapping was off by only 1–2 pixels between the two fields. The colocalized spots were calculated as a proportion of Cy5 spots for each frame and subsequently averaged for each oocyte. Then the oocyte values were pooled to calculate the mean ± s.e.m for presentation in text and figures. For quantification of the random colocalization, Cy5 frames of the encoded condition were rotated 90° from center and merged with the corresponding (non-rotated) Cy3 images.

## Statistical analysis

All values are presented as mean ± s.e.m. To determine statistical significance Student's *t*-test (two-tailed distribution; two-sample equal variance) was performed. The threshold for significance was p=0.05.

## EST database analysis

To classify the proteins that are expressed in *Xenopus laevis* ooctyes, we started with an annotated, non-normalized cDNA library as expressed sequence tags (ESTs) for *X. laevis* oocytes in stages V and VI, constructed by Blumberg et. al (*Blumberg et al., 1991*; *Hawley et al., 1995*). Each of the ESTs were then queried against *Xenopus laevis*'s Unigene database that contains 31306 sequence clusters (Build #94) (*Pontius et al., 2003*; *Boguski and Schuler, 1995*; *Schuler, 1997*). We identified 1305 unique mRNA sequences that were expressed in stage V/VI *X. laevis* oocytes. The open reading frame (ORF) was identified and transcribed from these mRNA sequences and the protein sequence was used to predict the number of transmembrane (TM) helices using TMHMM 2.0 (*Krogh et al., 2001*). TMHMM algorithm is currently the best performing algorithm and it correctly predicts 97–98% of the transmembrane helices. We would like to point out that that our goal was not to correctly predict transmembrane helices but to classify the protein as soluble or membrane proteins. We classified all proteins with more than one predicted transmembrane segment as a membrane protein. With this method we found 43 membrane proteins whose mRNA sequence showed a TAG stop codon. We call attention to the possibility that of 18 protein sequences with a single predicted transmembrane helix, the N-terminally situated TM segment could be a signal peptide that may or may not get cleaved post-translationally. Hence, our bioinformatics analysis might overestimate the number of membrane proteins expressed in *Xenopus laevis* oocytes by a small amount.

## Acknowledgements

CAA is supported by NIH/NIGMS (GM106569), an American Heart Association Established Investigator (A22180002) and is a member of the Membrane Protein Structural Dynamics Consortium, which is funded by NIH/NIGMS (GM087519). JLR and RC are supported by NIH/NIGMS (R00GM101016; GM120260) and the Roy J Carver Trust Young Investigator Award. We thank Dr Maria Spies for comments on the manuscript and Dr Julien Sebag for providing access to Spectramax i3. Further, we thank Nathan D Rossen, Hadiatou Sow and Yasmin Bou Karim for technical support. Mass spectra were obtained by Vic Parcell of the HRMS Facility at the University of Iowa.

## Additional information

### Funding

| Funder | Grant reference number | Author |
| --- | --- | --- |
| National Institutes of Health | GM106569 | Lilia Leisle |
| National Institute of General Medical Sciences | GM120260 | Janice L Robertson Rahul Chadda |
| National Institutes of Health | GM106569 | Daniel T Infield Jason D Galpin Christopher A Ahern |
| American Heart Association | A22180002 | Jason D Galpin Christopher A Ahern |
| National Institutes of Health | GM087519 | Jason D Galpin Christopher A Ahern |
| National Institutes of Health | GM101016 | Janice L Robertson |

The funders had no role in study design, data collection and interpretation, or the decision to submit the work for publication.

## Author contributions

LL, Conception and design, Acquisition of data, Analysis and interpretation of data, Drafting or revising the article; RC, JLR, Conception and design, Analysis and interpretation of data, Drafting or revising the article; JDL, Conception and design, Acquisition of data, Contributed unpublished essential data or reagents; DTI, Acquisition of data, Drafting or revising the article; JDG, Acquisition of data, Contributed unpublished essential data or reagents; VK, Conception and design, Analysis and interpretation of data; CAA, Conception and design, Drafting or revising the article

## Author ORCIDs

Janice L Robertson, http://orcid.org/0000-0002-5499-9943
Christopher A Ahern, http://orcid.org/0000-0002-7975-2744

# Additional files

## Supplementary files

• Supplementary file 1. Membrane proteins carrying TAG as native stop codons identified from EST databases of *Xenopus laevis* oocytes stage V and VI. In total, *Xenopus laevis* oocytes of stage V and VI express 1300 genes (see Materials and methods) and 18% of those (234 genes) contain TAG as a stop codon. 43 (shown here) are predicted to be membrane proteins, roughly half of which are categorized as mitochondrial proteins or proteins with unknown function.

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
