## [Decision Letter]

Thank you for submitting your article "Cellular encoding of Cy dyes for single-molecule imaging" for consideration by *eLife*. Your article has been favorably evaluated by Richard Aldrich (Senior Editor) and three reviewers, one of whom is a member of our Board of Reviewing Editors. The reviewers have opted to remain anonymous.

The reviewers have discussed the reviews with one another and the Reviewing Editor has drafted this decision to help you prepare a revised submission.

Summary:

In the manuscript, Ahern and colleagues describe genetic encoding of cyanine dyes as non-canonical amino acids. Such encoding provides orthogonal fluorescent labeling of any protein in a living cell. The method, presented in considerable detail, is discussed in the context of performing single molecule fluorescence studies with particular emphasis on single-molecule FRET of membrane proteins. This first demonstration of incorporation of cyanine dyes as ncAAs in eukaryotic expression systems is significant because these dyes are some of the best single-molecule analysis of protein dynamics and should considerably improve the analysis of stoichiometry in protein complexes. The present study is of a broad and potentially general interest to diverse scientists. The manuscript is well organized, appropriately referenced and well written. All three reviewers were positive regarding potential publication of the manuscript, but raised a considerable number of concerns that should be addressed by the authors. These are largely focused on the lack of clarity in the manuscript as to how applicable the method is to possible practical questions and what are reservations and limitations.

Essential revisions:

In particular, an apparently confusing aspect of work is that the efficiency of tRNA acylation and the efficiency of the downstream events (including labeled amino acid incorporation into proteins and biological activity of the labeled proteins) were not separated. For example, Figure 2) would appear to suggest that there are significant differences in channel activity depending on the dye used. However, the point of these panels is that the extent of channel activity observed correlates with the extent to which the tRNA underwent successful ligation in vitro prior to its injection. Why weren't the full-length tRNAs purified prior to injection to control for this variable? Published protocols are available from a number of groups including (Blanchard et al. PNAS 2004). The reviewers feel that it is important to address this question and suggest that at the very least the experiment shown in Figure 1 is redone with acylated tRNA purification prior to cell-free translation to establish the efficiency with which labeled amino acids (compared to unlabeled) are incorporated into proteins. The results shown in other figures (particularly Figure 2) should then be explicitly discussed in terms of which processes contribute to the observed differences.

Additional comments (some of which may be related to the above mentioned issue) are compiled below.

*Reviewer 1:*

First, it seems that incorporation efficiency drops with the size of the dye, such that Cy5-FA and Cy3-based LD550-FA are incorporated with lower efficiency than Cy3-FA. For example, Figure 1 shows that in 2 out of 3 experiments little protein was made with incorporation of Cy5-FA. What kind of levels of incorporation would make the approach practical? If one wanted to conduct FRET experiments, what levels would be sufficient?

Second, it is unclear what fraction of the label remains fluorescent at the end of the biosynthetic process in an oocyte. This question should be directly investigated, particularly since the presented data suggest that the fraction is significantly less than 1. For example, in Figure 4, half of CLC0 dimers photo-bleach in a single step. Considering that the total number of observed particles is significantly higher than in the mock experiments, the observation suggests that a fraction of the dye molecules in the dimers are no longer fluorescent. Alternatively, it is possible that a significant fraction of CLC0 proteins are not properly assembled into dimers. Either of these possibilities would compromise experimental design. Similarly, only a small fraction of Cy5 labeled proteins in Figure 3 co-localized with Cy3 label. Since Cy3-FA is incorporated with higher efficiency, one would expect that most of CLC0 dimers would contain either 2 Cy3-s or Cy3/Cy5. However, Cy5-only particles seem to dominate in Figure 3, right panel.

Finally, I was confused about molecules that photo-bleach in more than 2 steps. Are they higher-order assemblies? What fraction of total molecules do they constitute? In other words, if we did not know that CLC0 were a dimer, could we determine its stoichiometry from these experiments? If not, what are the main challenges that need to be resolved to make this method applicable in straightforward way?

*Reviewer 2:*

The authors should address several questions about the approach and results to help the reader to understand how well it may work for various membrane proteins.

1) Why is Cy3 fluorescence so much thinner a ring at the outer end of the fatter GFP ring in Figure 1?

2) How effective is the suppression compared to expression of the normal (wildtype) protein? E.g. in the case of ClC, as shown in Figure 2, how much current is generated if wildtype cRNA is injected in the same quantity? It is also not very clear how much the expression level was reduced (especially for the Na channel and the cell-free system), which makes it difficult to evaluate how useful this method can be on other poorly expressed channels. Authors should discuss factors that may contribute to lower expression levels in the most optimal case of suppression (e.g. with Phe). The manuscript did not address the relationship between the incorporation efficiency and the amount CyX-tRNA used. What are the concentrations that lead to maximum suppression of nonsense mutation? How does the incorporation efficiency change with different concentrations of cRNA and/or CyX-tRNA?

3) The cold ligation temperature does not seem to be suitable for cells. Will the increased susceptibility to hydrolysis be a problem even after the CyX-tRNA is injected (e.g. when the oocytes are incubated at 16°C)?

4) While injecting nonacylated tRNA with CLC-0 E166TAG is said to yield no functional channels, the reversal potential of CLC-0 E166TAG with no AA is still at -30mV. Is it possible that there was some expression? Additionally, it is surprising that having a bulky dye in the middle of the open pore did not generate any blocking effect, while it is shown in the literature that many E166 mutants of CLC-0 (E166 mutants) can be blocked by fatty acids and amphilic blockers much more easily than the WT. Can the authors provide more evidence to show that it is indeed Cl^-^ current carried by CLC-0 E166TAG incorporated with CyX? Does the property of the pore change in comparison to the Phe incorporation alone (such as blocker affinity)?

5) To estimate random colocalization the authors rotate the image by 90 degrees. Could they also use the spatial resolution of their microscope and a calculation of chance overlap at the experimental densities to obtain a second estimate?

6) In Figure 3 Cy3 spots vary greatly in brightness. A) Does this mean that the dim ones are single channels and bright ones are clusters? B) Is this the highest density obtainable (would higher expression mean a bigger difference between encoded and mock)?

7) In Figure 4 some mock spots have 2 bleaching steps. Is this compatible with explanation for what mock may be (e.g. tRNAs, etc.)?

8) In Figure 4 the encoded spots include ones with more than 2 bleaching steps that are attributed to ClC clusters. Is there any patch clamp data or other data that supports clustering of channels? Bar graphs only show up to 2 steps: what is the relative frequency of higher step spots?

9) One prediction for co-incorporation of Cy3 and Cy5 is that the dimmest spots with single Cy3 bleach steps should have only a single Cy5 bleach step. Is this the case? Alternatively, if only Cy3 were incorporated, the dimmest spots should have 2 steps and there should be no single steps. It would be good to show if one or both of these expectations is satisfied.

10) Any speculation on the effects of Cy-tRNA labeling on the native TAG codon of other channels? Will that be an issue when using this technique for single molecule imaging/subunit counting?

*Reviewer 3:*

1) It is difficult to discern from the data in Figure 1 whether the extent of Cy3 and Cy5 labeling is quantitative, which one would presume to be the case. While stoichiometric labeling with Cy5 seems to have occurred based on the image shown, this does not appear to be the case for Cy3. Was the image chosen scaled differently?

2) The site of labeling chosen for ClC-0 is somewhat odd given that the chosen position is a transmembrane residue in the ion permeation pathway (Results and Discussion, fourth paragraph). Although this is evidently not the case based on the functional data shown in Figure 2, I would imagine the bulky dye addition would cause issues in buried positions. Indeed, there does seem to be some variance in the different dyes employed (Figure 2). The authors may wish to clarify further why they chose this position versus one that is solvent exposed.

3) None of the dye structures shown in Scheme 1 are correct. They should either be corrected or, as their inclusion is not germane to the findings of the paper, removed.

---

## [Author Response]

*Summary:*

*In the manuscript, Ahern and colleagues describe genetic encoding of cyanine dyes as non-canonical amino acids. Such encoding provides orthogonal fluorescent labeling of any protein in a living cell. The method, presented in considerable detail, is discussed in the context of performing single molecule fluorescence studies with particular emphasis on single-molecule FRET of membrane proteins. This first demonstration of incorporation of cyanine dyes as ncAAs in eukaryotic expression systems is significant because these dyes are some of the best single-molecule analysis of protein dynamics and should considerably improve the analysis of stoichiometry in protein complexes. The present study is of a broad and potentially general interest to diverse scientists. The manuscript is well organized, appropriately referenced and well written. All three reviewers were positive regarding potential publication of the manuscript, but raised a considerable number of concerns that should be addressed by the authors. These are largely focused on the lack of clarity in the manuscript as to how applicable the method is to possible practical questions and what are reservations and limitations.*

We thank the reviewers for their constructive feedback, and in particular for highlighting the confusion about the application of this approach for single-molecule studies. We have now amended the manuscript to make it clear how this method is currently useful. First, the incorporation of a bright, organic dye into the protein backbone in live cells provides a new strategy for in vivo fluorescent labeling. While we observe that the fluorescent yield decreases after introduction into oocytes, optimization of this approach with oxygen scavenger systems or photostable dyes will likely address this issue in the next generation of experiments. Still, at this time, the incorporation of a cyanine dye into membrane proteins in cells even at low levels, introduces new opportunities for single-molecule studies. This includes measurements of binding equilibrium of fluorescently labeled ligands/toxins/proteins using co-localization microscopy, low background single-molecule voltage clamp fluorometry of ion channels, and, once established in other cell expression systems, can provide an ideal method for obtaining long-lived diffusional tracking of membrane proteins within membranes. While the fluorescent yield needs to be increased to make this method useful for subunit counting and single-molecule FRET, the data we present here demonstrates that it is possible to incorporate large cyanine dyes into proteins via the ribosome and thus reports a significant advance in site-specific fluorescent labeling of proteins in living cells.

*Essential revisions:*

*In particular, an apparently confusing aspect of work is that the efficiency of tRNA acylation and the efficiency of the downstream events (including labeled amino acid incorporation into proteins and biological activity of the labeled proteins) were not separated. For example, Figure 2) would appear to suggest that there are significant differences in channel activity depending on the dye used. However, the point of these panels is that the extent of channel activity observed correlates with the extent to which the tRNA underwent successful ligation in vitro prior to its injection. Why weren't the full-length tRNAs purified prior to injection to control for this variable? Published protocols are available from a number of groups including (Blanchard et al. PNAS 2004). The reviewers feel that it is important to address this question and suggest that at the very least the experiment shown in Figure 1 is redone with acylated tRNA purification prior to cell-free translation to establish the efficiency with which labeled amino acids (compared to unlabeled) are incorporated into proteins. The results shown in other figures (particularly Figure 2) should then be explicitly discussed in terms of which processes contribute to the observed differences.*

In response to the Essential revisions we have analyzed the tRNA ligations of Cy3- and Cy5-ncAA by HPLC as suggested by the reviews. This data is now included as Figure 2—figure supplement 1 and shows that the ligation efficiencies of Phe and the two noncanonical amino acid fluorophores are similar and high (>90%). Thus, the decreased expression of proteins suppressed with Phe compared to tRNA harboring the larger dyes, such as Cy5 and LD550, may be due to intrinsic encoding efficiencies as suggested by the reviewers. This is now discussed in the resubmitted manuscript and is motivation for future studies. However, subsequent attempts to reclaim HPLC purified tRNA resulted in poor yields (<10%), making a quantitative comparison of purified vs. unpurified tRNAs infeasible. This matter will be eventually addressed with large-scale tRNA productions but this possibility exceeds our current production capacity.

We note that the ClC-0 channels produced with unpurified tRNA ligations with Phe or any of the encoded Cy dyes display identical gating behavior, but reduced expression at the plasma membrane. Thus, the increased size of the Cy-ncAA does not impact function (at this site) but appears to lower the expression. The inefficient nature of nonsense suppression is relevant to the application of the technique and is now discussed in the context of in vitro and oocyte expression systems. In particular, in the in vitro protein reaction, rescue of the NanoLuc Amber nonsense codon appears to be very low in comparison to WT expression, ~7,000-fold for Phe and ~13,000-fold for Cy3, respectively (Figure 2, Figure 2—figure supplement 2). In the case of the oocyte expression system, new data is provided on the rescue versus WT expression of ClC-0 indicating that suppressed channels are roughly 100-fold lower in their expression (Figure 3—figure supplement 2). We therefore highlight in the manuscript the remarkable result that these dyes can be encoded in the eukaryotic ribosome, and make clear that their application is intended for single-molecule studies, thus only requiring very small amounts of encoded protein. We hope that this discussion of suppression efficiency helps to clarify the utility of the approach. Its use for macroscopic FRET, for instance, would be challenging in its current iteration. Further, we discuss how encoding rates may influence the overall expression of a protein harboring the Cy-ncAA and how the determination of the fluorescent fraction of encoded dyes remains an important challenge prior to their systematic use to study distances by single molecule FRET or protein complex stoichiometry.

We have removed the in vitro protein synthesis data concerning GFP152*TAG* rescue with THG73 tRNA carrying Cy3 or Cy5. In the original manuscript, rescue of GFP152*TAG* was investigated by confocal imaging of single beads. We have since begun to more systematically quantify the in vitro rescue of GFP152*TAG* by bulk fluorometry bead imaging. The data here indicate that rescue is low and not statistically significant above background. Thus, the individual confocal images of GFP on single beads is not representative of the bulk population, which we believe, is a more accurate representation of the overall rescue. The in vitro system remains valid however, as we have further confirmed the encoding of Cy3-ncAA within the NanoLuc-Amber construct using pyrrolysine tRNA. This data also indicates that the rescue of NanoLuc-Amber is well below WT NanoLuc expression, see above. This may suggest that the in vitro system is less efficient compared to the oocyte but this possibility, as well as the general optimization of in vitro encoding conditions will require additional investigation. The main thrust of the manuscript remains the demonstration the encoding of the Cy dyes in membrane proteins in live cells. Therefore, the changes above in regards to cell free protein synthesis do not alter the overall conclusions of the study, but further highlight the need to optimize this method when working with new protein systems, something that is expected whenever working with a new technique.

Another aspect of the original submission that led to unintended confusion is the observation of photobleaching steps beyond that expected for the two-steps for a dimer. This could result from expression of the channels beyond single-molecule levels (over-expression), or from clustering of ClC-0 channels in the oocyte plasma-membrane. Clustering has been reported previously for expressed ClC-0 in the oocyte (Bauer et al., 1991). Both of these conditions precludes the direct examination of ClC-0 channel stoichiometry, although the former could be titrated down to achieve single-molecule expression. In either case, the appearance of higher order bleaching step events has not been seen in control conditions and, hence, provides some of the strongest evidence for encoding of the Cy-ncAA in the ClC-0 channel. Therefore, this remains an important observation.

However, in order to further clarify this issue for the reader, as well as the possible experimental applications of the approach in its current iteration (a related topic), the following paragraph that outlines possible causes for this result (i.e. spots that bleach with > 2 steps) has been added to the relevant section in the Results and Discussion:

“There remain a number of issues that must be overcome before using this method for stoichiometry measurements. First, the basis for the observation of spots that bleach with > 2 steps merits further investigation. […] Thus, while additional effort is needed to maximize the possible applications for encoding Cy-ncAA, there are a handful of applications at this stage. Further, we anticipate that ultimately it will be applicable to mammalian cell lines once approaches for tRNA delivery have been established.”

Additional changes to the manuscript motivated by reviewer requests include new data depicting that rescued ClC-0 E166*TAG* currents are indeed mediated by the chloride channel ClC-0 (Figure 3—figure supplement 1). Moreover, we show the amplitude of rescued ClC-0 currents as a function of tRNA amount injected into oocytes (Figure 3—figure supplement 2). Also included are a discussion of suppression of endogenous stop sites and a new associated bioinformatics analysis of TAG codon abundance as native stop codon in oocyte membrane proteins.

*Additional comments (some of which may be related to the above mentioned issue) are compiled below.*

*Reviewer 1:*

*First, it seems that incorporation efficiency drops with the size of the dye, such that Cy5-FA and Cy3-based LD550-FA are incorporated with lower efficiency than Cy3-FA. For example, Figure 1 shows that in 2 out of 3 experiments little protein was made with incorporation of Cy5-FA. What kind of levels of incorporation would make the approach practical? If one wanted to conduct FRET experiments, what levels would be sufficient?*

In order to address the role of encoding ability versus ligation efficiency we have analyzed our tRNA acylation reactions by HPLC (Figure 2—figure supplement 1), also please see above. This new data is consistent with that in the original Figure 2 and shows that under the current conditions, 4°C for 2 hours, both Cy3 and Cy5 have very high, and similar, ligation efficiencies. Thus, as suggested by the reviewer, the difference in the expression of proteins with Cy3 or Cy5 (or LD550) at the same site is likely to be due to reduced rates of intrinsic encoding.

We speculate that the bigger dyes encode more slowly due (possibly) to slower transit rates in the ribosome, and are more likely to have aborted peptide synthesis events. This may explain the apparent low encoding rates in in vitro protein synthesis reactions for proteins that otherwise show very good expression, such as GFP. This possibility is currently being examined in ongoing experiments that are beyond the scope of the current manuscript but could shed new light on the role of ribosomal pausing sites and encoding efficiency.

In terms of the general low expression profile of proteins harboring Cy-ncAA, this is common when using nonsense suppression with in vitro acylated, orthogonal tRNA. Although, as stated above, the larger Cy dyes do appear to be reaching the size limit for encoding abilities but this possibility merits a more deliberate structure-encoding investigation. However, the goal of this project at the outset was not to achieve macroscopic FRET but to encode a non-canonical fluorophore with proven utility for single-molecule studies. The Cy-ncAA introduced here meets this criterion. In the context of single molecules, the reduced expression level is not necessarily a shortcoming. And while it is clear that, for the ClC-0 example, the channels show a 100’s-fold decrease in expression levels, this is still millions of channels in the oocyte, thus likely to be enough to yield meaningful single-molecule signals, either in situ in the oocyte plasma membrane or after purification. Further, our group has recently published a complementary method using known endoplasmic retention signals and split intein sequences to enable the suppression of multiple stop sites within a protein complex or reading frame (Lueck et al., 2016). Lastly, the pyrrolysine tRNA used in Figure 2 has been shown previously to be amenable to anticodon mutagenesis to allow for the specific encoding at dedicated stop sites i.e. TAG versus TGA (Ambrogelly et al., 2007). Thus, while some key steps are in place for the eventual application of single-molecule FRET, as is the case with any new approach, protein specific technical issues will no doubt need to be overcome.

*Second, it is unclear what fraction of the label remains fluorescent at the end of the biosynthetic process in an oocyte. This question should be directly investigated, particularly since the presented data suggest that the fraction is significantly less than 1.*

This is likely to be the case under the current conditions for the synthesis and encoding, i.e. the fraction of fluorescent dye is below 1. However, the data obtained in the oocyte system clearly indicate that it is greater than 0. The HPLC analysis of the ligated tRNA as suggested by the reviewer is quite helpful in understanding this phenomenon (Figure 2—figure supplement 1). Here, the data show that after synthesis and coupling to tRNA, the Cy-ncAA remain fluorescent and the apparent loss of fluorescence occurs within the oocyte and during downstream encoding events. Future experiments with oxygen scavengers and more stable dyes will help in this regard. Nonetheless, the interpretation of the data, which is now more clearly emphasized in the resubmitted manuscript, is that it overwhelmingly supports the notion that the dyes can be encoded in a eukaryotic ribosome. This single observation alone is significant. Thus, of all the potential problems that one may have anticipated for encoding a large dye for single-molecule imaging, the ribosome is surprisingly tolerant. Systematically assessing the fraction of dark dyes and how this can be further improved is an important part of ongoing efforts.

*For example, in Figure 4, half of CLC0 dimers photo-bleach in a single step. Considering that the total number of observed particles is significantly higher than in the mock experiments, the observation suggests that a fraction of the dye molecules in the dimers are no longer fluorescent. Alternatively, it is possible that a significant fraction of CLC0 proteins are not properly assembled into dimers. Either of these possibilities would compromise experimental design. Similarly, only a small fraction of Cy5 labeled proteins in Figure 3 co-localized with Cy3 label. Since Cy3-FA is incorporated with higher efficiency, one would expect that most of CLC0 dimers would contain either 2 Cy3-s or Cy3/Cy5. However, Cy5-only particles seem to dominate in Figure 3, right panel.*

As noted above, we agree with the premise of the reviewer that a fraction of the dyes is likely to be non-fluorescent. New HPLC data provided in the resubmission indicates that the Cy dyes, once on tRNA, is still completely fluorescent (Figure 2—figure supplement 1). Thus the loss of fluorescence occurs after injection into the oocyte and during subsequent encoding. How this occurs and, more importantly, how to remedy this issue during encoding is an active area of investigation. This may involve the application of additional ‘self-healing’ dyes, like the ‘encodable’ LD550. In reference to the specific questions on predicted outcomes of colocalization and bleaching steps, these issues are associated with the use of ClC-0 and are therefore likely to be further complicated by the channel clustering (mentioned above). However, the abundance of single bleaching steps for Cy3 is unlikely due to the possibility that ClC-0 is not properly assembled into dimers given that the encoding site, E166, is not near the protein interface of the dimer, and substitutions at this position in homologue proteins crystalized as dimers (Dutzler et al., 2003; Lobet & Dutzler, 2006). We also direct the reviewer to the two new paragraphs above.

*Finally, I was confused about molecules that photo-bleach in more than 2 steps. Are they higher-order assemblies? What fraction of total molecules do they constitute? In other words, if we did not know that CLC0 were a dimer, could we determine its stoichiometry from these experiments? If not, what are the main challenges that need to be resolved to make this method applicable in straightforward way?*

Please see the response to the Essential revisions as well as the two new paragraphs above outlining challenges and applications of the method.

*Reviewer 2:*

*The authors should address several questions about the approach and results to help the reader to understand how well it may work for various membrane proteins.*

*1) Why is Cy3 fluorescence so much thinner a ring at the outer end of the fatter GFP ring in Figure 1?*

The GFP Y152*TAG* rescue data has been removed from the resubmitted manuscript, see above.

*2) How effective is the suppression compared to expression of the normal (wildtype) protein? E.g. in the case of ClC, as shown in Figure 2, how much current is generated if wildtype cRNA is injected in the same quantity? It is also not very clear how much the expression level was reduced (especially for the Na channel and the cell-free system), which makes it difficult to evaluate how useful this method can be on other poorly expressed channels. Authors should discuss factors that may contribute to lower expression levels in the most optimal case of suppression (e.g. with Phe).*

The relevant point here to consider is that the that loss of expression is considerable, but in terms of using this application for single-molecule studies, suppression is not a limitation because one only needs a couple of molecules to be incorporated into the membrane to be visualized by TIRF microscopy. Specifically, the magnitude of reduction in yield upon nonsense suppression in oocytes is significant and highly variable in our experience at various positions within a target cRNA as well as between channel types. In the case of ClC-0 E166TAG, the expression of the rescued channels with Phe is at least *100-fold* less than the wild-type ClC-0 channel (Figure 3—figure supplement 2). This estimate was obtained by comparing the reduction of the injected cRNA needed to achieve a similar expression profile as a rescued ClC-0 channel. In our hands, other channel proteins fair better, while others do worse.

To address the specific point of the reviewer, there are multiple factors that impact the overall efficiency of suppression yield and these are now discussed in the resubmitted manuscript (Dougherty & Van Arnam, 2014; Leisle et al., 2015). For one, there is competition with the endogenous translation terminator, release factor 1 (RF-1), for binding to TAG codons. Second, the in vitro transcribed and folded tRNA is not modified while native tRNAs are and it is not known whether modifications might still occur in the cell after injection into oocytes. Further, a proportion of the synthetized tRNA molecules might not be properly folded. Consequently, the in vitro synthesized, orthogonal tRNA might display a lower translation efficiency compared to native tRNAs. Third, amino acid hydrolysis from the tRNA is pH and temperature dependent and might occur during incubation of the translation reactions in cell-free or in oocytes prior to forming a complex with the endogenous elongation factors (Peacock et al., 2014). However, the latter is not supported by the fact that membrane protein expression generally improves over the course of days.

*The manuscript did not address the relationship between the incorporation efficiency and the amount CyX-tRNA used. What are the concentrations that lead to maximum suppression of nonsense mutation? How does the incorporation efficiency change with different concentrations of cRNA and/or CyX-tRNA?*

To maximize yields the amount of injected aminoacyl-tRNA is critical. We now submit a new figure that describes the effect of tRNA concentration, in this example Val-tRNA, on subsequent rescue of ClC-0 E166*TAG*, (Figure 3—figure supplement 2). In our experience, the amount of cRNA is less important and is varied only to reduce read-through effects of the TAG codon, if necessary.

*3) The cold ligation temperature does not seem to be suitable for cells. Will the increased susceptibility to hydrolysis be a problem even after the CyX-tRNA is injected (e.g. when the oocytes are incubated at 16°C)?*

The ligation of the pCA-Cy dye conjugate to tRNA occurs in vitro, prior to injection into the oocyte. The use of lower ligation temperature (4°C) allows for the production of high tRNA acylation efficiency, (Figure 2, Figure 2—figure supplement 1). However, once injected into the oocyte we cannot rule out that hydrolysis of the amino acid from the tRNA in some form is not a factor in the subsequent expression profiles of various channel proteins and specific sites. This is not unique to the Cy-ncAA described here and given the wide-spread success of nonsense suppression generally in the oocyte this appears to be unlikely. For instance, the expression of rescued proteins will increase over the course of 24-48 hours after the injection of target cRNA and acylated tRNA. This suggests that the tRNA are present in the acylated form over this period. It is possible that the injected tRNA subsequently forms a stable complex with an endogenous elongation factors, which are abundant in eukaryotes, a possibility that would significantly reduce in-cell hydrolysis (Peacock et al., 2014).

*4) While injecting nonacylated tRNA with CLC-0 E166TAG is said to yield no functional channels, the reversal potential of CLC-0 E166TAG with no AA is still at -30mV. Is it possible that there was some expression? Additionally, it is surprising that having a bulky dye in the middle of the open pore did not generate any blocking effect, while it is shown in the literature that many E166 mutants of CLC-0 (E166 mutants) can be blocked by fatty acids and amphilic blockers much more easily than the WT. Can the authors provide more evidence to show that it is indeed Cl^-^ current carried by CLC-0 E166TAG incorporated with CyX? Does the property of the pore change in comparison to the Phe incorporation alone (such as blocker affinity)?*

In the main text we show that upon co-injection of ClC-0 E166*TAG* cRNA and non-acylated tRNA into oocytes, the stop-codon read-through at position 166 is minimal (Figure 3). The specific incorporation of Phe, Cy3- and Cy5-ncAA produced between about 6-11 times larger currents at +80 mV than the read-through control. As noted by the reviewer, even with a minimal expression of a chloride channel in the oocyte plasma membrane one would expect the membrane potential to approach the chloride equilibrium potential. Therefore, we compared oocytes simply injected with water, non-acylated tRNA + ClC-0 E166*TAG* cRNA or Phe-tRNA + ClC-0 E166*TAG* cRNA and examined the subsequent reversal potential and current amplitude. Moreover, we perfused these oocytes with iodide, a ClC-0 blocker (Pusch et al., 1995; Ludewig et al., 1997; Zifarelli & Pusch, 2007) to provide evidence that the measured currents are carried by ClC-0 channels. Endogenous chloride conductances to the oocyte have a I^-^ > Cl^-^ selectivity (Weber, 1999), thus water injected oocytes are expected to increase their conductance upon exchange of Cl^-^ to I^-^. Both expectations have been met by new data in the resubmitted manuscript (Figure 3—figure supplement 1). This is consistent with the notion that the encoded Cy dyes are well tolerated functionally. Also, available structures of open pore mutants of EcClC-1, a ClC-0 homologue (EcClC-1 E148Q and EcClC-1 E148A by Dutzler et al., 2003; EcClC-1 S107A/E148Q/Y445A by Lobet & Dutzler, 2006) show that substitutions at this position cause the side chain to reorient towards the solvent accessible extracellular space. Given that the Cy chromophore in this example is attached to a para-amino-Phe moiety by a six carbon linker, it is likely that this would allow the fluorophore to protrude well into the extracellular space, away from the permeation pathway.

*5) To estimate random colocalization the authors rotate the image by 90 degrees. Could they also use the spatial resolution of their microscope and a calculation of chance overlap at the experimental densities to obtain a second estimate?*

While the reviewer makes a valid suggestion, image rotation to estimate incidental (chance) co-localization is an equally established method, see Figure 2F in Dunn et al., 2011 and Figure 10B in Wang et al., 2001. The strength of image rotation being that it relies on the same optical resolution, and spot-density enabling estimation of a negative control, thus providing an important internal control. Hence, performing the analysis suggested by the reviewer involving additional Monte Carlo simulations to increase ‘n’ would be redundant given the current data set.

*6) In Figure 3 Cy3 spots vary greatly in brightness. A) Does this mean that the dim ones are single channels and bright ones are clusters? B) Is this the highest density obtainable (would higher expression mean a bigger difference between encoded and mock)?*

A) The reviewer makes an important observation. One possibility is that the brighter spots are generated by clusters of multiple ClC-0 CyX containing channels. However, spot brightness does not always correlate with the number of fluorophores (unless they are closely neighboring spots) because the field of view is not equally illuminated, thus the spots off center are excited with a steadily decreasing amount of energy.

B) We agree that one would expect a higher expression to increase the difference to the mock reaction. Efforts to improve encoding efficiency are underway.

*7) In Figure 4 some mock spots have 2 bleaching steps. Is this compatible with explanation for what mock may be (e.g. tRNAs, etc.)?*

The data show that 2-step bleaching events in the mock condition are rare, but consistent (and expected) with the ‘mock’ condition given the known cellular architecture of the oocyte plasma membrane. Specifically, in the text, we speculate that ‘The fluorescent spots observed in the mock condition may be ascribed to non-encoded Cy-ncAA-tRNA or hydrolyzed Cy-ncAA that may have escaped the cytosol during peeling the vitelline membrane (a prerequisite to TIRF imaging) or that are trapped within the small cytosolic volume of the microvilli.’ The latter condition may account for 2-step bleaching events.

*8) In Figure 4 the encoded spots include ones with more than 2 bleaching steps that are attributed to ClC clusters. Is there any patch clamp data or other data that supports clustering of channels? Bar graphs only show up to 2 steps: what is the relative frequency of higher step spots?*

Please see general comments above on the multi-step bleaching events. Also, clustering of multiple ClC-0 channels in the oocyte membrane has been observed previously by labs performing excised patch experiments (see discussion in Bauer et al., 1991). For the Cy3 fluorescence, 3+ step events have been observed in ~21% of the total Cy3 spots and for Cy5 in ~13% of the total Cy5 spots.

*9) One prediction for co-incorporation of Cy3 and Cy5 is that the dimmest spots with single Cy3 bleach steps should have only a single Cy5 bleach step. Is this the case? Alternatively, if only Cy3 were incorporated, the dimmest spots should have 2 steps and there should be no single steps. It would be good to show if one or both of these expectations is satisfied.*

We agree with the premise of this comment but this analysis is not easily applied to the current ClC-0 dataset. For one, channel clustering, a possibility that was not anticipated at the onset, would allow for multiple channel dimers within a single spot. This is further complicated by the possibility of non-fluorescence dyes. However, barring these factors, both expectations would be met. This is an important point, however, and experiments are currently being planned for other channel types that may yield more predictable analytical outcomes.

*10) Any speculation on the effects of Cy-tRNA labeling on the native TAG codon of other channels? Will that be an issue when using this technique for single molecule imaging/subunit counting?*

It is a valid concern but we believe that the effect, if existent at all, will be very negligible for the following reasons: First, the expression level of endogenous proteins is significantly smaller than of an overexpressed protein. Second, from analyses of EST databases for *Xenopus laevis* oocytes of stage V/VI, we know that about 1,300 genes are expressed and 18% of those (234 genes) contain TAG as a stop codon. Thereof only 43 are actual membrane proteins with more than half of them being either mitochondrial proteins or proteins with unknown function ([Supplementary-material SD1-data]). Thus, the number of possible candidates is very low. Third and most important, when a stop codon is over-read in eukaryotic cells, the poly-A tail of the mRNA translates to a poly-Lys chain which forms a signal for ubiquitin-mediated protein degradation. So none of these proteins is likely to find its way to the plasma-membrane.

*Reviewer 3:*

*1) It is difficult to discern from the data in Figure 1 whether the extent of Cy3 and Cy5 labeling is quantitative, which one would presume to be the case. While stoichiometric labeling with Cy5 seems to have occurred based on the image shown, this does not appear to be the case for Cy3. Was the image chosen scaled differently?*

All images were scaled to the same settings. The data has been removed from the original manuscript because subsequent attempts to quantify the rescue of GFP Y152*TAG* with Cy3- or Cy5-ncAAs did not yield statistically significant differences compared to the negative control. The rescue was poor and imaging of single beads is not representative of the whole population.

*2) The site of labeling chosen for ClC-0 is somewhat odd given that the chosen position is a transmembrane residue in the ion permeation pathway (Results and Discussion, fourth paragraph). Although this is evidently not the case based on the functional data shown in Figure 2, I would imagine the bulky dye addition would cause issues in buried positions. Indeed, there does seem to be some variance in the different dyes employed (Figure 2). The authors may wish to clarify further why they chose this position versus one that is solvent exposed.*

At first glance, as the reviewer suggests, it would appear that E166 may seem an odd choice to encode large fluorescent dyes. However, this gets back to the reasoning for using ClC-0 and the E166 site. For instance, extensive previous mutagenesis indicates that E166 shows broad tolerance for many types of side-chains. Thus, the bulky cyanine dyes are unlikely to be in the middle of the open pore. This possibility is further supported by available crystal structures of the open pore mutants of CLC proteins that have been resolved (Dutzler et al., 2003; Lobet & Dutzler, 2006), suggesting that substitutions at this position cause the side chains to orient towards the extracellular space. We now discuss this information in the text.

*3) None of the dye structures shown in Scheme 1 are correct. They should either be corrected or, as their inclusion is not germane to the findings of the paper, removed.*

Structures shown in the manuscript are based on detailed information provided by the manufacturers and are consistent with mass spectrometry (Figure 1—figure supplement 2–Figure 1—figure supplement 4).